# Measurement properties of the one-minute sit-to-stand test in children and adolescents with cystic fibrosis: A multicenter randomized cross-over trial

Yann Combret[1,2,3]*, Fairuz Boujibar[4], Charlotte Gennari[5], Clément Medrinal[3,6,7], Sophie Sicinski[8], Tristan Bonnevie[9,10], Francis-Edouard Gravier[9,10], Muriel Laurans[5], Christophe Marguet[8,11], Pascal Le Roux[1], Bouchra Lamia[3,10,12], Guillaume Prieur[2,3,10], Grégory Reychler[2,13]

1 Pediatric Department, Le Havre Hospital, Le Havre, Normandie, France, 2 Research and Clinical Experimentation Institute (IREC), Pulmonology, ORL and Dermatology, Louvain Catholic University, Brussels, Brussels Capital Region, Belgium, 3 Pulmonology Department, Le Havre Hospital, Le Havre, Normandie, France, 4 INSERM U1096, UNIROUEN, Normandie Univ, Rouen University Hospital, Rouen, Normandie, France, 5 Cystic Fibrosis Department, Pediatric Section, Caen University Hospital, Caen, Normandie, France, 6 Erphan Department, UVSQ, Paris-Saclay University, Versailles, Île-de-France, France, 7 Saint Michel School of Physiotherapy, Paris-Saclay University, Paris, Île-de-France, France, 8 Pediatric Cystic Fibrosis Department, Pediatric Section, Rouen University Hospital, Rouen, Normandie, France, 9 ADIR Association, Rouen University Hospital, Rouen, Normandie, France, 10 UPRES EA3830 - GRHV, Institute for Research and Innovation in Biomedicine (IRIB), UNIROUEN, Rouen University Hospital, Rouen, Normandie, France, 11 Pediatric Respiratory Disease and Allergy Unit, UNIROUEN, Inserm EA 2656, Rouen University Hospital, Rouen, Normandie, France, 12 Intensive Care Unit, Respiratory Section, Rouen University Hospital, Rouen, Normandie, France, 13 Department of Physical Medicine and Rehabilitation, Saint-Luc University Clinics, Brussels, Brussels Capital Region, Belgium

* yann.combret@gmail.com

**Data Availability Statement:** All relevant data are within the manuscript and its Supporting information files.

## Abstract

### Background

Functional exercise capacity assessment is recommended in children with cystic fibrosis (CF). The six-minute walk test (6MWT) is a valid evaluation of exercise capacity but can be technically complex. Inversely, the sit-to-stand test (STST) is a simple method to evaluate exercise capacity, and is validated in healthy children and adults with CF. This study aimed to evaluate STST measurement properties in children and adolescents with CF.

### Methods

In this multicenter study, children with CF (6 to 18 years) performed two iterations of both the STST and the 6MWT in a randomized order. Criterion validity was determined by assessing correlations between STST repetitions and 6MWT distance (6MWD). Intra-rater reliability, test-retest repeatability, mean bias and limits of agreement were also assessed. Relationships with other outcomes (i.e. respiratory and quadriceps muscle strength) and cardio-respiratory responses were analysed for both tests.

**Funding:** The authors received no specific funding for this work.

**Competing interests:** I have read the journal's policy and the authors of this manuscript have the following competing interests. Prof. Bouchra Lamia reports expertise activities from Novartis, Chiesi, and from Astra Zeneca, non-financial support from Philips Respironics, grants from Lowenstein, grants from Bayer, and expertise activies from Elivie, all outside the submitted work. Mr. Tristan Bonnevie reports grants from Fisher & Paykel, outside the submitted work. All the other authors have no competing interests to declare. This does not alter our adherence to PLOS ONE policies on sharing data and materials.

**Abbreviations:** 6MWD, six-minute walking distance; 6MWT, six-minute walk test; 6MWW, six-minute walking work; CF, cystic fibrosis; CFQ-R, cystic fibrosis questionnaire revised; CPET, cardio-pulmonary exercise testing; DBP, diastolic blood pressure; FEV1, forced expiratory volume in one second; FVC, forced vital capacity; HR, heart rate; HRQOL, heahth-related quality of life; ICC, intra-class correlation coefficient; MEP, maximal expiratory pressure; MIP, maximal inspiratory pressure; PEF, peak expiratory flow; QS, quadriceps strength; RR, respiratory rate; SBP, systolic blood pressure; SpO2, pulsed oxygen saturation; STST, sit-to-stand test; STSTW, sit-to-stand test work; TV, tidal volume; VO$_2$peak, maximal oxygen consumption at peak exercise.

## Results

Thirty-six children with CF were included (mean age 12.0 ±3.5 years and FEV1 95.8 ±25.0%). On average, 39.6 ±10.5 repetitions were performed during the STST and mean 6MWD was 596.0 ±102.6 meters. STST number of repetitions was significantly correlated with 6MWD (r = 0.48; p<0.01). Both tests had very good intra-rater reliability (ICC$_{STST}$ = 0.91 (95%CI 0.76–0.96) and ICC$_{6MWT}$ = 0.94 (95%CI 0.85–0.97)), and a significant test-retest learning effect. The number of STST repetitions was not correlated with quadriceps or respiratory muscle strength test, and the STST induced fewer cardio-respiratory responses than the 6MWT.

## Conclusions

The STST is an easy-to-use functional test with moderate criterion validity when compared to the 6MWT in children with CF, probably because both tests measure different components of functional exercise capacity. The STST is useful when the 6MWT is unfeasible, however further investigations are required to explore the clinical implications of STST results in children with CF.

## Clinical trial registration

NCT03069625.

## Introduction

Functional exercise capacity assessment is of primary importance in individuals with cystic fibrosis (CF) [1]. Recent guidelines have recommended the use of cardiopulmonary exercise testing (CPET) and the six-minute walk test (6MWT) in children with CF aged 10 years and older [2]. Unfortunately, CPET requires equipment and technical skills that are not available everywhere and requires further adaptation in children aged <10 years [2, 3]. The 6MWT is a reliable test in children with CF [4]. The distance walked during the test (6MWD) has been found to be closely associated with VO$_2$peak (maximal oxygen consumption at peak exercise) in children with severe obstruction, and 6-minute walking work (6MWW: 6MWD x body-weight (kg)) has been shown to be associated with VO$_2$peak in children with a moderate decline in lung function [5, 6]. The 6MWT is widely used in clinical practice but since it requires a 30-metre long corridor, its use is limited in general wards or outpatient environments [7].

In contrast, the one-minute sit-to-stand test (STST) requires neither a lot of technical equipment nor a lot of space. It is a simple and reliable functional exercise test that has been validated in adults with CF [8]. The number of STST repetitions is associated with VO$_2$peak, maximal power and the physical functioning scale of the Cystic Fibrosis Questionnaire (CFQ-R) in adults with moderate-to-severe CF [8]. Derived STST measures (PowerSTST or STST x bodyweight = STSTW) have also shown strong associations with VO$_2$peak, maximal power and quadriceps muscle strength in adults with CF [9, 10].

The STST has recently been validated in healthy children but, to date, it has not been used to assess children with CF [11]. Based on its ease of use for both general practitioners and children, we hypothesized that the STST would be of interest as an accessible means of functional

exercise capacity testing in children with CF in modest clinical settings. Our study aimed to: (1) explore STST criterion validity to measure functional exercise capacity compared to the 6MWT in children with CF; (2) evaluate STST intra-rater reliability, test-retest reliability, mean bias and limits of agreement; and (3) analyse correlations with other outcomes previously identified as accounting for functional exercise capacity (respiratory and quadriceps muscle strength, nutritional status, etc.).

## Materials and methods

### Participants

Children with CF were assessed for eligibility in one of 3 French CF Centers in either Le Havre, Rouen or Caen. In order to be included, they had to have a diagnosis of CF (sweat chloride > 60mmol/L), be aged between 6 and 18 years and to have been clinically stable (absence of hospitalizations or exacerbations requiring intravenous or oral antibiotics) for at least 4 weeks prior to the study. Ethical approval was obtained from the Comité de Protection des Personnes Nord-Ouest III. Participants and their parents provided informed signed consent. This study was registered on ClinicalTrial Database (NCT03069625) and followed the CONSORT guidelines.

### Study design

Participants performed both the STST and 6MWT in the same day in a cross-over design and the order of the tests was randomized. The randomization sequence was computer-generated by a person not involved in the study using a random block size on a 1:1 ratio and was sent to each centre at the beginning of the study. Participants performed two rounds of each test to avoid learning effects and all the tests were supervized by the same investigator in each centre (YC and GP in Le Havre, FB and SC in Rouen and CG in Caen). All the investigators were senior physiotherapists, experienced (at least 5 years) in the assessment of muscle and functional capacities in individuals with CF. Furthermore, a preliminary visit was made to the Rouen and Caen centers by the principal investigator (YC) prior to the first inclusion to homogenize testing procedures. Participants first performed STST1 and 6MWT1 (according to randomization) within a 30-minute delay. Participants then performed STST2 and 6MWT2 after a 30-minute break and within a 30-minute delay. The same tests (STST1 and STST2; 6MWT1 and 6MWT2) were then performed within a 1 hour delay.

### The STST

The STST was performed using a vertically adjustable chair without arm-rests to accomodate the variability in participants' leg length. The chair height was set to ensure 90˚ of knee flexion during the test. Participants were instructed to stand-up completely and sit-down at their own pace for 1 minute, without using their arms as leverage or support, as previously described [12]. They were allowed to perform several sit-to-stand movements before the test to familiarize themselves with the procedure. Standardized information was given before the test and no verbal encouragements were provided during the test. Participants were allowed to take rests during the test. The number of completed STST cycles was recorded by the investigators. As a significant learning effect has been reported in adults with CF, two STST were performed [8]. Heart rate (HR), pulsed oxygen saturation (SpO2), systolic and diastolic blood pressure (SBP, DBP), and respiratory rate (RR) were measured during a 1-min sitting rest period before and after each STST (53NOP portable monitor, WelchAllyn). Dyspnea and lower limb fatigue were also recorded using the modified Borg scale.

### The 6MWT

The 6MWT was conducted according to ATS/ERS guidelines [7]. Participants were asked to walk as far as possible at their own pace. As for the STST, standardized information was provided before testing and no verbal encouragements were given. The investigator used standardized sentences to inform participants of the time remaining every minute. The total distance covered (6MWD) was recorded by the investigators to the nearest meter. HR, SpO2, SBP, DBP, RR, dyspnea and lower limb fatigue were also recorded during a 1-min sitting rest period before and after each 6MWT.

### Study procedure and outcomes

Participant characteristics were recorded upon arrival, including lung function if the last spirometry was more than 6-months old. Otherwise, lung function was evaluated according to ATS/ERS guidelines prior to the muscle and functional testing [13]. Then, before the functional testing, respiratory muscle strength was assessed using maximal inspiratory and expiratory pressures (MIP and MEP), measured with an electronic manometer (MicroRPM, Eolys). In accordance with ATS/ERS guidelines, participants stood in an upright position during the test and wore a nose clip. Standardized information and verbal encouragements were provided. The best of three inspiratory and expiratory manoeuvres that varied by less than 10%, with a 60 second rest period between each measurement, were recorded and used for analysis [14]. Moreover, quadriceps strength (QS) was also assessed prior to functional testing using an isometric measurement of the maximal voluntary contraction of the dominant limb with a hand-held dynamometer (Microfet2, Biometrics). Participants were seated on a quadriceps chair with both lower limbs positioned in 90° of hip and knee flexion. The dominant leg was fixed using a non-elastic belt and their arms were crossed over their chest. The dynamometer was placed between the belt and the leg. Participants were asked to extend their leg as hard as possible against the belt. After two repetitions for familiarization, 3 trials were conducted, with a 60 second rest between each measurement, and the best used in the analysis. Standardized information was provided before the test and verbal encouragements were provided during the test [15].

Functional testing was then undertaken according to the protocol described above and health-related quality of life (HRQOL) was evaluated using the French version of the CFQ-R during the rest period between two iterations of the field tests [16]. For children under 14 years of age, a parent also completed the parental version of the CFQ-R.

The number of repetitions achieved during the two STST and the distance achieved during the two 6MWT were recorded. The best performances of each test were used to evaluate criterion validity and correlations with other outcomes. According to the latest international statement, exercise testing (using either CPET or 6MWT) should be undertaken yearly in all individuals with CF from the age of 10 [2]. We therefore also aimed to investigate STST criterion validity separately in the children ≤ 10 years of age and those aged 11–18 years. Cardiorespiratory adaptations during the best performance of each test were also compared.

### Statistical analysis

The sample size estimation showed that 36 participants were necessary to detect at least a moderate correlation with a correlation coefficient of 0.45 between the STST and 6MWT performances, with 80% power and an alpha level of 0.05 [17, 18]. The final sample size aimed for was 40 since it was estimated that there would be an attrition rate of approximately 10%, based on experience in using these clinical tests with children: the 2x2 STST and 6MWT protocols together, for example, lasted for at least 2 hours and had to be completed within the same day.

Reasonable allowance was also made for unexpected factors such as the inability of a participant to complete all of the tests or the withdrawal of consent following inclusion in the study.

Data are expressed as numbers and percentages for categorical variables and as means (±standard deviations) or medians (interquartile range) for quantitative variables and the range of participants' values (minimum-maximum) for the sample characteristics. The normality of the distribution for each variable was assessed using the Kolmogorov-Smirnov test. All statistical analyzes were performed using Graph Pad Prism 8. Criterion validity of the STST to measure functional exercise capacity was explored by measuring the correlation between STST performance and the 6MWD derived from the reference test performance (i.e. 6MWT), using Spearman or Pearson correlation coefficients in the complete study sample, in children ≤10 years of age and in children aged 11–18 years. Relationships with other outcomes were explored using Spearman or Pearson correlation coefficients in the complete study sample, in children ≤10 years of age and in children aged 11–18 years. Correlation coefficients were interpreted as follows: 0 to 0.19 = very weak correlation, 0.20 to 0.39 = weak correlation, 0.40 to 0.59 = moderate correlation, 0.60 to 0.79 = strong correlation and 0.80 to 1.0 = very strong correlation [18]. Cardio-respiratory adaptation and patient-reported outcomes during the tests were compared using paired t-tests or Wilcoxon Signed Rank tests. Intra class correlation coefficients (ICC) were calculated using a random effects model for coherence of mean measures to verify intra-rater reliability between the first and the second test for both STST and 6MWT. ICC values were interpreted as follows: <0.5 = poor reliability; 0.5–0.75 = moderate reliability; 0.75–0.90 = good reliability; >0.90 = very good reliability [19]. Repeatability between the first and second iterations of both tests was calculated using a paired Student T test. The Bland-Altman method was used to determine bias and limits of agreement [20]. The significance level was set at p<0.05.

## Results

Forty-nine children with CF were considered as potentially eligible into the study. Of these, thirty-six were included (n = 11 in Le Havre, n = 14 in Rouen and n = 11 in Caen) between August 2017 and August 2019 and completed the entire study (Fig 1).

Participants characteristics are detailed in Table 1. Briefly, participants were 12.0 ±3.5 years old and presented mild bronchial obstruction (FEV1 95.8 ±25.0%) (Table 1).

The average number of repetitions performed in the second STST was 39.6 ±10.5 and the distance walked in the second 6MWT was 596.0 ±102.6 metres (Table 2).

The number of STST repetitions was correlated with 6MWD (r = 0.48; p<0.01) in the entire sample of children. Criterion validity of the STST with the 6MWT was also moderate in children ≤ 10 years of age (r = 0.54; p<0.05), and in children > 10 years of age (r = 0.43; p<0.05) (Fig 2).

The number of STST repetitions was neither correlated to QS (r = -0.01; p = 0.99), MIP (r = 0.31; p = 0.07), MEP (r = 0.32, p = 0.06) nor lung function (S1 Table). STST was not correlated with HRQOL items, except for child-estimated social functioning (r = 0.37; p = 0.03) (S2 Table). Inversely, 6MWD correlated with every muscle strength assessment (r = 0.48; r = 0.54 and r = 0.46; p<0.01 for MIP, MEP and QS respectively) and with lung function (r = 0.59; r = 0.55 and r = 0.58; p<0.01 for FVC, FEV1 and PEF respectively) (S1 Table). 6MWD was associated with several HRQOL items including child physical functioning (S2 Table).

Baseline cardio-respiratory characteristics were similar before both tests. Concomitant increases in both HR and RR results indicated that the 6MWT produced a greater cardio-respiratory demand than the STST. Children also reported greater dyspnoea and more lower limb fatigue during the 6MWT (Table 3).

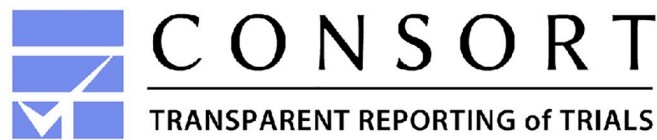

## CONSORT 2010 Flow Diagram

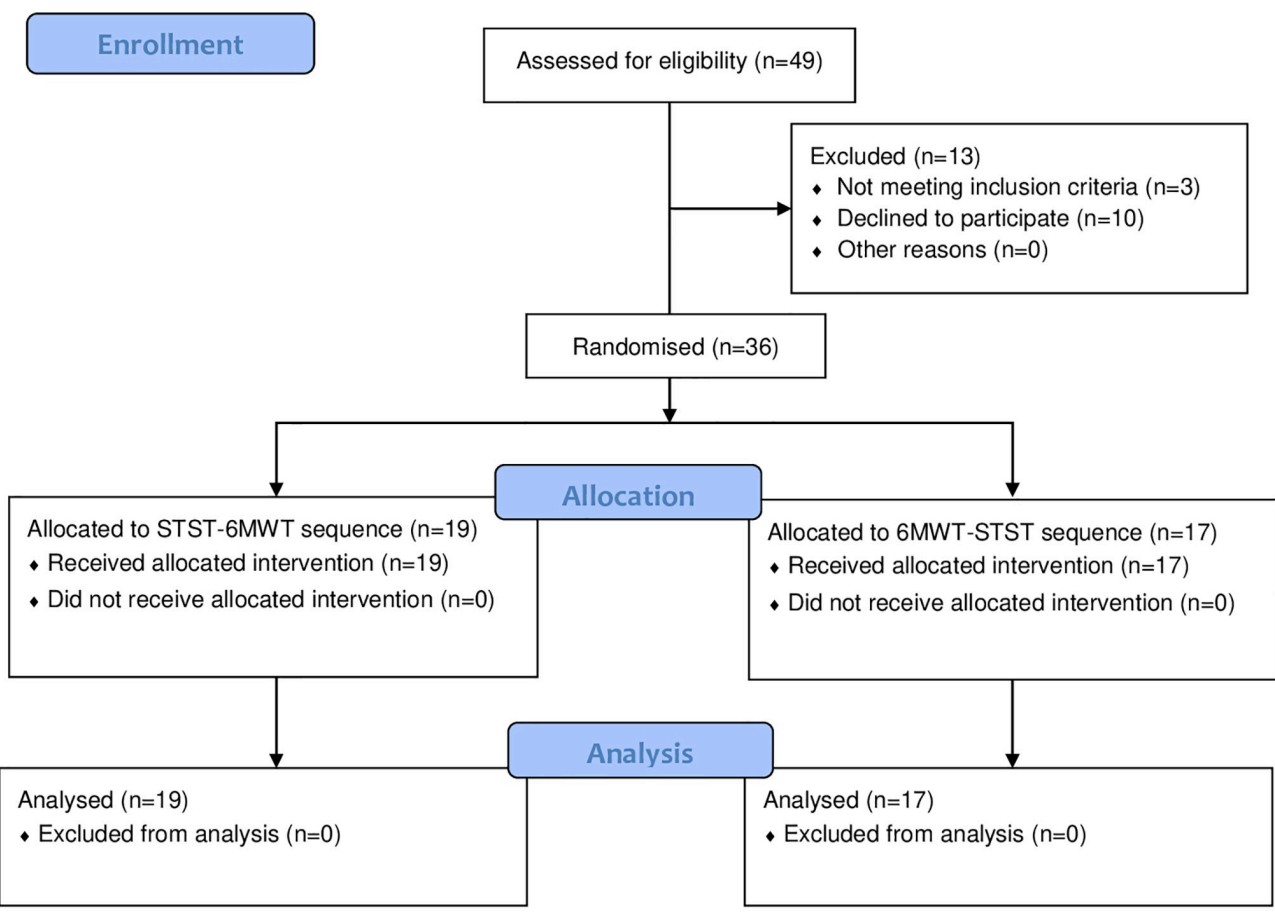

**Fig 1. Flow diagram illustrating patient participation (CONSORT).**

Changes in HR during both tests correlated with participant height (r = 0.38 and r = 0.42; p = 0.02 for STST and 6MWT respectively). Delta HR during the STST correlated with lung function (r = 0.36, r = 0.46 and r = 0.50; p<0.05 with FVC, FEV1 and PEF respectively) and with child-reported HRQOL total scores (r = 0.46; p<0.01) whereas delta HR during 6MWT correlated with FEV1 and PEF (r = 0.35 and r = 0.51; p = 0.04 and p<0.01 respectively).

A significant learning effect was observed for both tests (STST mean difference = 3.1 repetitions (95%CI = 1.3–4.9); 6MWD mean difference = 22.9 meters (95%CI = 7.9–37.9); p<0.01).

**Table 1. Study population characteristics.**

| Outcomes | | Children ≤10 years of age (n = 14) | Children aged 11–18 years (n = 22) | All children (n = 36) | Complete range |
|---|---|---|---|---|---|
| *Anthropometric characteristics and genotype* | | | | | |
| Male / Female (n) | | 9/5 | 5/17 | 14 / 22 | - |
| Age (years) | | 8.2 ± 1.6 | 14.4 ± 1.9 | 12.0 ± 3.5 | 6–17 |
| Height (cm) | | 125.7 ± 11.0 | 158.4 ± 11.1 | 145.7 ± 19.5 | 110–192 |
| Weight (kg) | | 24.4 ± 6.4 | 46.1 ± 9.2 | 37.6 ± 13.4 | 17–76.1 |
| BMI (kg.m$^2$) | | 15.2 ± 1.7 | 18.2 ± 1.6 | 17.0 ± 2.2 | 11.9–21.8 |
| Homozygote F508del, n (%) | | 5 (35.7) | 10 (45.5) | 15 (41.6) | - |
| Heterozygote F508del, n (%) | | 7 (50) | 12 (54.5) | 19 (52.8) | - |
| Other, n (%) | | 2 (14.3) | 0 (0) | 2 (5.6) | - |
| *Lung function* | | | | | |
| FVC | (L) | 1.8 ± 0.7 | 3.0 ± 0.8 | 2.5 ± 1.0 | 0.9–4.9 |
| | (% predicted value) | 101.4 ± 23.4 | 93.0 ± 18.8 | 96.3 ± 20.8 | 61–154 |
| FEV1 | (L) | 1.5 ± 0.5 | 2.5 ± 0.8 | 2.1 ± 0.9 | 0.8–4.2 |
| | (% predicted value) | 102.7 ± 22.9 | 91.4 ± 25.8 | 95.8 ± 25.0 | 49–137 |
| FEV1/FVC (%) | | 86.9 ± 6.2 | 84.0 ± 10.0 | 85.1 ± 8.8 | 66–97 |
| PEF | (L.min) | 3.4 ± 0.8 | 5.5 ± 1.6 | 4.7 ± 1.7 | 1.7–9.1 |
| | (% predicted value) | 93.4 ± 18.5 | 86.2 ± 21.7 | 89.0 ± 20.6 | 42–119 |

Data are shown as either means (±standard deviations) or medians (interquartile ranges), and ranges (minimum-maximum).

BMI: body mass index; CF: cystic fibrosis; FEV1: forced expiratory volume 1 second; FVC: forced vital capacity.

Mean bias was 3.1 repetitions (95% limits of agreement -7.3–13.5) between STST1 and STST2 and 22.9m (95% limits of agreement -63.8–109.6) between 6MWT1 and 6MWT2 (Fig 3). Both tests showed very good intra-rater reliability with both ICCs above 0.9 (ICC$_{STST}$ = 0.91 (95%CI 0.76–0.96) and ICC$_{6MWT}$ = 0.94 (95%CI 0.85–0.97)).

**Table 2. Muscle and functional outcomes.**

| Outcomes | | Children ≤10 years of age (n = 14) | Children aged 11–18 years (n = 22) | All children (n = 36) | Complete Range |
|---|---|---|---|---|---|
| *Muscle strength and functional exercise capacity* | | | | | |
| MIP | (cmH$_2$O) | 60.3 ± 20.3 | 74.9 ± 31.9 | 69.2 ± 28.5 | 18–149 |
| | (% predicted value) | 112.8 ± 37.9 | 100.5 ± 43.8 | 105.4 ± 40.7 | 23.7–227.4 |
| MEP | (cmH$_2$O) | 57.0 ± 24.4 | 75.2 ± 26.3 | 68.1 ± 26.8 | 22–140 |
| | (% predicted value) | 91.4 ± 39.9 | 71.5 ± 24.2 | 77.2 ± 27.8 | 30.6–138.4 |
| Quadriceps strength | (N) | 138.4 ± 95.0 | 273.9 ± 86.4 | 221.2 ± 108.0 | 30–416 |
| STST | (number of repetitions) | 38.3 ± 10.9 | 40.5 ± 10.5 | 39.6 ± 10.5 | 24–76 |
| 6MWD | (m) | 573.6 ± 120.9 | 610.3 ± 89.2 | 596.0 ± 102.6 | 415–845 |
| | (% predicted value) | 88.7 ± 17.0 | 81.4 ± 13.6 | 84.3 ± 13.8 | 62.7–110.8 |
| *Health-related quality of life questionnaire scores* | | | | | |
| Child-completed total score | | 73.7 ± 13.4 | 76.1 ± 11.2 | 75.2 ± 11.9 | 41.3–93.5 |
| Parent-completed total score | | 74.9 ± 12.4 | 72.7 ± 10.1 | 74.2 ± 11.5 | 51.1–91.5 |
| CF exacerbations within the last year | | 2 (1.2) | 2.5 (3) | 2 (2) | 0–10 |
| CF hospitalizations within the last year | | 0 (0.3) | 0 (1) | 0 (0.8) | 0–2 |

Data are shown as either means (±standard deviations) or medians (interquartile ranges), and ranges (minimum-maximum).

CF: cystic fibrosis; MEP: maximal expiratory pressure; MIP: maximal inspiratory pressure; STST: sit-to-stand test; 6MWT: six-minute walking test.

**2-A**

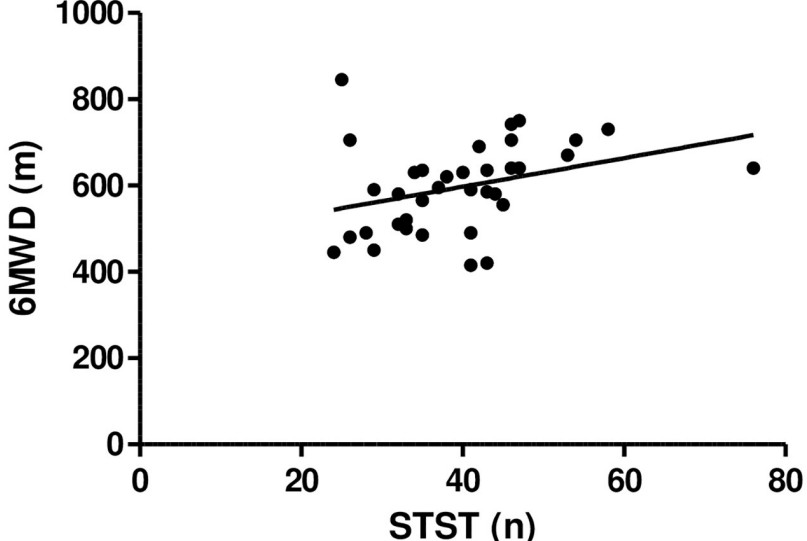

**2-B**

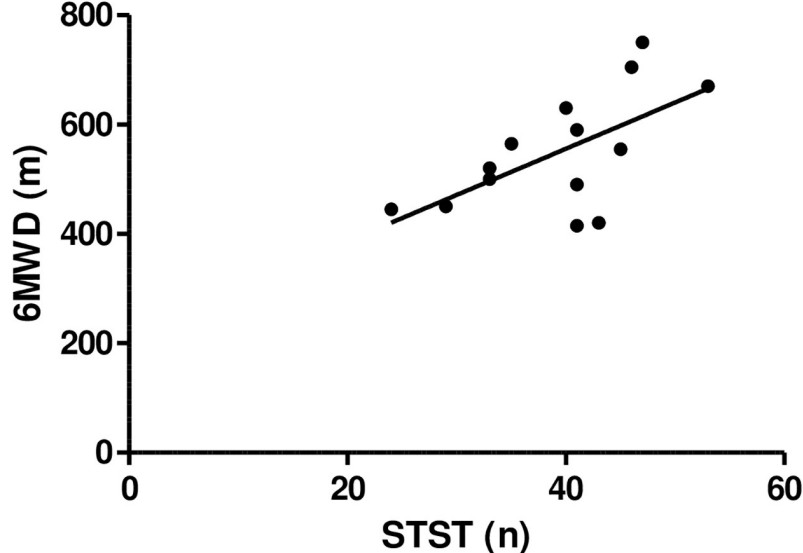

**Fig 2. Relationship between the number (n) of movement repetitions performed during the STST and the distance (m) walked during the 6MWT; A: 36 children with CF aged 6 to 18 years (r = 0.48; p<0.01) and B: 14 children with CF aged 6 to 10 years (r = 0.54; p<0.05).**

## Discussion

This study showed that the STST had moderate criterion validity for the measurement of functional exercise capacity when compared to the 6MWT (r = 0.49) in the entire study group of

**Table 3. Pre- and post-exercise test cardio-respiratory response comparisons for the STST and 6MWT in children with CF.**

| Outcomes | STST (n = 36) | | 6MWT (n = 36) | | p-value (Pre-exercise) | Change scores (ΔPre-post exercise) | | p-value (Change scores) |
|---|---|---|---|---|---|---|---|---|
| | *Pre-exercise* | *Post-exercise* | *Pre-exercise* | *Post-exercise* | | *STST* | *6MWT* | |
| HR (beats.min) | 95.8±14.4 | 116.3±19.9 | 96.4±17.8 | 126.4±23.9 | 0.71 | 20.5±18.4 | 29.9±20.3 | <0.01 |
| RR (breaths.min) | 20.6±4.9 | 26.6±5.1 | 20.1±4.5 | 30.3±9.7 | 0.55 | 6.1±5.2 | 10.2±8.7 | <0.01 |
| $SpO_2$ (%) | 96.8±2.0 | 96.6±1.9 | 97.1±2.0 | 96.4±2.8 | 0.22 | -0.2±1.4 | -0.7±2.4 | 0.18 |
| SBP (mmHg) | 106.8±10.6 | 112.7±12.0 | 106.9±10.1 | 115.4±13.7 | 0.82 | 5.8±7.9 | 8.4±9.9 | 0.21 |
| DBP (mmHg) | 64.7±8.4 | 69.2±10.2 | 64.7±9.8 | 70.7±10.3 | 0.52 | 4.4±9.6 | 6.0±7.8 | 0.44 |
| Dyspnoea (0–10) | 0(0) | 2(2.5) | 0(0) | 3 (2) | 0.34 | 2(2) | 3(2) | <0.01 |
| Leg fatigue (0–10) | 0(0.9) | 1(2) | 0(0.5) | 2(3.4) | 0.35 | 1(1) | 2(2) | 0.04 |

DBP: diastolic blood pressure; HR: heart rate; RR: respiratory rate; SBP: systolic blood pressure; SpO2: transcutaneous oxygen saturation; STST: sit-to-stand test; 6MWT: six-minute walking test. Values are shown as means±SD (with 95% CI) or as medians (interquartile ranges).

children with CF and in children ≤10 years of age (r = 0.54). The STST was not correlated to other outcomes whereas the 6MWT was correlated with muscle strength (respiratory muscles and quadriceps) and HRQOL. A significant learning effect and a very good intra-rater reliability were reported for both tests. The STST induced a significantly lower cardio-respiratory response, dyspnoea and leg fatigue than the 6MWT.

This study is the first to investigate STST validity to measure functional exercise capacity compared to the 6MWT in children with CF. A similar significant correlation between the number of STST repetitions and the 6MWD has been described in young healthy adults (18–25 years) (r = 0.64) [21]. Gruet et al. reported positive correlations between cardiorespiratory variables during the STST, CPET and 6MWT in adults with CF, although an association between the STST and 6MWD was not reported [10]. A correlation (r = 0.63) between STST performance and CPET-derived VO2peak has also been reported for adults with CF undergoing pulmonary rehabilitation [9]. None of the children in this study underwent CPET before or during our study due to equipment unavailability. This moderate correlation can be explained by the different domains of physical capacity investigated by both tests, and the corresponding differences in energy expenditure [22, 23]. The sit to stand movement is relevant for activities of daily living and is repeated frequently throughout the day by both adults and children. The lack of ability of the one-minute STST to predict the distance covered during a 6MWT can be attributed to the functional differences between the tests, as well as differences in energy utilization between walking for 6-minutes and standing repeatedly from a chair for 1-minute. Interestingly, a longer variation of the one-minute STST (the 3-minute STST) induced a similar functional response to that of the 6MWT in patients with COPD, leading the authors to conclude that the tests were equivalent [24]. Nevertheless, it should be noted that the 3-minute STST induced a higher degree of leg fatigue than the 6MWT, reinforcing the suggestion that the tests have different functional demands and should not be used interchangeably. The STST remains useful if the 6MWT cannot be performed.

In contrast, the STST has previously been shown to provide an adequate reflection of anaerobic capacities and consequently of quadriceps strength, even though there are conflicting results in the literature [10, 25–27]. In the present study, quadriceps strength was not correlated with the number of STST repetitions. We suggest three reasons for this lack of correlation. Firstly, other factors, such as balance or quadriceps endurance (two components not measured in this study), also contribute to the sit-to-stand movement, thus quadriceps strength alone cannot fully explain STST performance [28, 29]. Secondly, although quadriceps

**3-A**

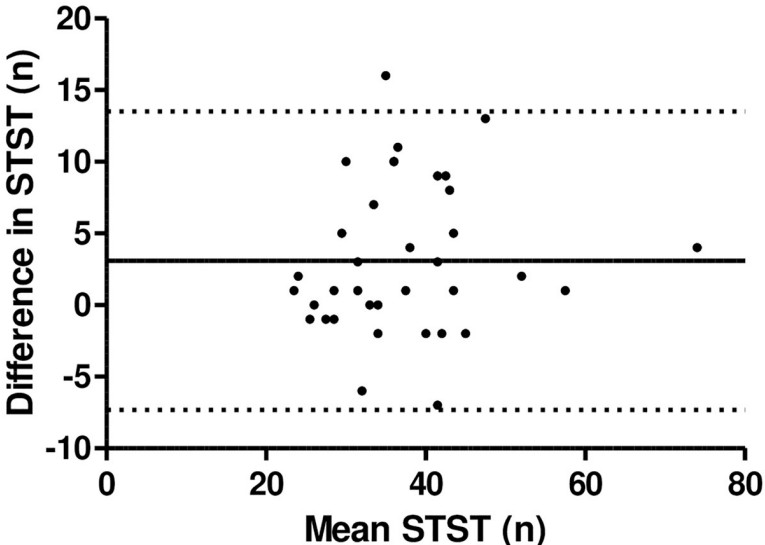

**3-B**

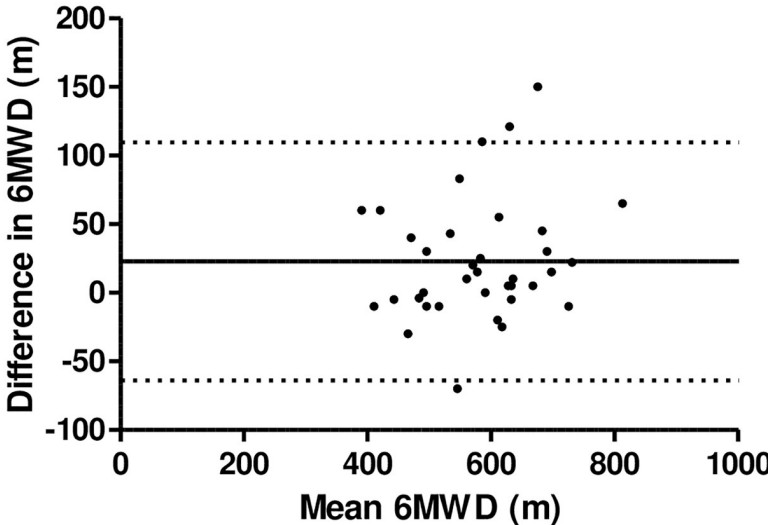

**Fig 3. Two scatterplots showing: A: Repeatability of the delta and the average number (n) of movement repetitions performed during the STST and; B: Repeatability of the delta and the average distance walked (m) during the 6MWT by 36 children with CF aged 6–18 years (Bland-Altman method).**

strength testing using fixed handheld dynamometry has been shown to be reliable and accurate for patients with chronic respiratory insufficiency, its reliability in children with CF has been little studied [30, 31]. Finally, the results for both quadriceps strength and STST performance may have been affected by the age and height of the children in this study. This effect could have been reduced by normalising the results by leg length: evidence suggests that the

product of STST performance and body weight produce higher correlations with lower limb strength [10, 26]. Finally, a small trend for a higher correlation between STST and 6MWT performance in children ≤ 10 years of age, compared to older children (> 10 years of age) was observed in this study (r = 0.54 vs r = 0.43, respectively). Interestingly, a sex effect on both STST and 6MWT performance has previously been described; and the sex ratio was unequally distributed between those subgroups (64 vs 23% of boys between the children ≤ 10 years of age and >10 years of age, respectively) [32, 33]. A higher heterogeneity in both tests performances in the group of children ≤ 10 years of age could thereby have led to a higher correlation coefficient [18]. Therefore, the STST could be a promising tool for exercise testing in young children with CF, but confirmation in a larger sample of participants is needed.

The mean number of STST repetitions performed in the present study was around 40 (±10). A previous study reported that healthy children (mean age 12 years) performed an average of 42 STST repetitions [11]. Adults with CF at the end of a 3-week pulmonary rehabilitation program performed 67 repetitions in a previous study but it would be expected that even with CF, at the end of a rehabilitation program adults would have higher exercise capacities than children [8]. For example, adults with CF, moderate-to-severe obstruction (FEV1 59%) and a low physical activity index only performed an average of 40 repetitions [10]. Data regarding the number of expected STST repetitions or predictive equations for healthy children are lacking in the literature. However, STST performance in these different studies may be explained by age-related differences in height and weight as well as the level of airway obstruction and physical activity level.

Surprisingly, the average 6MWD of 596m in this study was considerably lower than the average distance of 742m reported previously in a sample of children of similar age and lung function [6]. One possible explanation is that only a few children in our sample had previously performed the 6MWT. Some CF centers (including those participating in this study) perform exercise testing in the case of exercise-related symptoms and not as part of routine evaluations [3, 34]. Another possible explanation is that the children had a lack of intrinsic motivation to perform four tests in a row. Also, the children included in our study had relatively low expiratory muscle strength (77% of the predicted value) whereas MIP was within normal values. A relationship between MEP and 6MWD (r = 0.60) has previously been described in children with CF and may explain the low 6MWD [35]. A recently described cohort of adults with CF (MEP 77% predicted value) similarly reported that those with lower MEP had poorer functional exercise capacity [36]. However, explanations underlying this observation are still unclear since other studies have reported differing results [37, 38].

End-exercise HR during the 6MWT reported in studies including healthy children and children with CF were considerably higher than in the present study with respectively 158 and 154 beats/min compared to 126 beats/min [33, 35]. Another recent study of healthy children reported a higher peak HR during the STST compared to our study (139 compared to 116 beats/min) [11]. Despite these differences we found that the 6MWT induced a greater cardiorespiratory response than the STST. Gruet et al found a similar higher peak HR in adults with CF when they underwent the 6MWT compared to the STST [10]. This difference in cardiorespiratory adaptation, which has previously been demonstrated in healthy adults performing the same tests, may be due to the different durations of each test [27]. Greater adaptation could be expected for a six-minute walk compared to a task like standing repeatedly from a chair for only one minute. Furthermore, walking could involve a larger number of muscles, inducing a faster HR and a higher level of leg fatigue and dyspnea. These differences further strengthen the hypothesis that these tests measure different aspects of functional capacity. Inversely, higher RRs have been reported during the STST than the CPET in adults with CF, although participants were also found to have lower tidal volumes (TV) [9]. TV was not

measured in our study but one explanation may be that in children with only mild obstruction and during a one minute effort, TV increased, rather than RR [39].

The simplicity of the STST along with its low cardio-respiratory demand makes it an attractice tool for the measurment of exercise performance in children, especially in the case of limited resources, but also as a means to prepare these children for the further exercise testing they will encounter during adolescence. The STST is a simple assessment, well-suited to the needs and behaviours of young children since it requires only one chair, takes just a few minutes to perform and can be conducted either in a clinical or a home setting. Assessment of children in their own home reduces the stress and costs associated with unnecessary hospital visits as well as helping to reduce the spread of nosocomial infections [40]. The test can also be useful for children with CF for whom CPET or the 6MWT are impractical or impossible. A minimum of two STST should be performed, however, to allow children to become familiarized with the test method. Furthermore, it is important to note that the STST does not measure the same aspects of functional capacity as the 6MWT, and that its discriminative validity as well as its clinical applications (e.g. the association with hospitalizations for CF or other important prognostic outcomes) have yet to be demonstrated.

## Limitations

There are several limitations to this study. First, the children included had a large age range. Although these population ages are consistent with the children with CF previously used to assess 6MWT, we also found evidence of considerable natural variation in functional exercise capacity related to aging and maturation. Secondly, few children had previously undertaken either the 6MWT or STST before this study. Future studies may wish to consider assessing the relationship between STST and CPET-derived outcomes in children with CF. Furthermore, STST should be conducted using more than two iterations to determine precisely the number of rounds needed to obtain the most reliable STST performance to assess functional exercise capacity.

## Conclusions

The results of this study suggest the STST measures different aspects of functional capacity than the 6MWT in children with CF. STST performance was moderately correlated with 6MWD, and the test induced a lower cardio-respiratory response. Neither quadriceps nor respiratory muscle strength were related to the number of STST repetitions, and the clinical value of the test is still to be proven. The simplicity of the STST for both children and practitioners, and the very good intra-rater reliability reported in this study, makes it a reasonable second option when 6MWT or CPET cannot be performed, but healthcare providers should be aware that these tests cannot be used interchangeably. Nonetheless, as both tests revealed a significant learning effect that should be taken to account, the STST could be helpful to reduce the duration of exercise testing.

## Supporting information

**S1 Table. Correlations between functional exercise capacity measured with the STST or the 6MWT and other clinically meaningful outcomes in 36 children with CF.**
(DOCX)

**S2 Table. Correlations between functional exercise capacity measured with the STST or the 6MWT and the health-related quality of life questionnaire (HRQOL) responses as**

**provided by the children with CF and by one of their parents for children under 14.**
(DOCX)

**S1 File. CONSORT 2010 checklist of information to include when reporting a randomised trial**\*.
(DOC)

**S2 File.**
(XLSX)

**S3 File.**
(DOCX)

**S4 File.**
(DOCX)

# Acknowledgments

The authors would like to thank the whole team of Le Havre, Caen and Rouen Cystic Fibrosis Centers for their support in this study. Authors would also like to thank the GHAHR Association (Le Havre, France) for their support, and Johanna Robertson and Jennifer Dandrea Palethorpe for language assistance and constructive criticism.

# Author Contributions

**Conceptualization:** Yann Combret, Fairuz Boujibar, Clément Medrinal, Tristan Bonnevie, Francis-Edouard Gravier, Pascal Le Roux, Bouchra Lamia, Guillaume Prieur, Grégory Reychler.

**Data curation:** Yann Combret, Fairuz Boujibar, Charlotte Gennari, Clément Medrinal, Sophie Sicinski, Muriel Laurans, Christophe Marguet, Pascal Le Roux, Guillaume Prieur.

**Formal analysis:** Yann Combret.

**Investigation:** Yann Combret, Charlotte Gennari, Sophie Sicinski, Francis-Edouard Gravier, Muriel Laurans, Guillaume Prieur, Grégory Reychler.

**Methodology:** Yann Combret, Charlotte Gennari, Clément Medrinal, Tristan Bonnevie, Francis-Edouard Gravier, Christophe Marguet, Bouchra Lamia, Guillaume Prieur, Grégory Reychler.

**Project administration:** Yann Combret, Bouchra Lamia, Grégory Reychler.

**Supervision:** Fairuz Boujibar, Tristan Bonnevie, Francis-Edouard Gravier, Muriel Laurans, Christophe Marguet, Pascal Le Roux, Bouchra Lamia, Guillaume Prieur, Grégory Reychler.

**Validation:** Yann Combret, Fairuz Boujibar, Charlotte Gennari, Clément Medrinal, Sophie Sicinski, Tristan Bonnevie, Francis-Edouard Gravier, Muriel Laurans, Christophe Marguet, Pascal Le Roux, Bouchra Lamia, Guillaume Prieur, Grégory Reychler.

**Writing – original draft:** Yann Combret.

**Writing – review & editing:** Yann Combret, Fairuz Boujibar, Charlotte Gennari, Clément Medrinal, Sophie Sicinski, Tristan Bonnevie, Francis-Edouard Gravier, Muriel Laurans, Christophe Marguet, Pascal Le Roux, Bouchra Lamia, Guillaume Prieur, Grégory Reychler.

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
