## [Decision Letter · Decision Letter 0]

16 Sep 2020

PONE-D-20-11696

Validation of the one-minute sit-to-stand test to measure functional exercise capacity in children and adolescents with cystic fibrosis: a multicenter randomized cross-over trial

PLOS ONE

Dear Dr. Combret,

Thank you for submitting your manuscript to PLOS ONE. After careful consideration, we feel that it has merit but does not fully meet PLOS ONE’s publication criteria as it currently stands. Therefore, we invite you to submit a revised version of the manuscript that addresses the points raised during the review process.

Major concern has been raised by the reviewers on the sample size and the methodology of the validation process. Clarification is required on several aspects of the indicators chosen. Consider if you can respond to the criticism satisfying the reviewers' requests or if thay are so fundamental (e-g- increase sample size) not to be fulfilled in a rieasonable time frame.

Please submit your revised manuscript by October 15 If you will need more time than this to complete your revisions, please reply to this message or contact the journal office at plosone@plos.org. Please include the following items when submitting your revised manuscript:

We look forward to receiving your revised manuscript.

Kind regards,

Andrea Martinuzzi

Academic Editor

PLOS ONE

Journal Requirements:

2.Thank you for stating the following in the Competing Interests section:

[I have read the journal's policy and the authors of this manuscript have the following competing interests:

Prof. Bouchra Lamia reports expertise activities from Novartis, Chiesi, and from Astra Zeneca, non-financial support from Philips Respironics, grants from Lowenstein, grants from Bayer, and expertise activies from Elivie, all outside the submitted work.

Mr. Tristan Bonnevie reports grants from Fisher & Paykel,  outside the submitted work.

All the other authors have no competing interests to declare. ].

Reviewers' comments:

Reviewer's Responses to Questions

**Comments to the Author**

1. Is the manuscript technically sound, and do the data support the conclusions?

Reviewer #1: Yes

Reviewer #2: Yes

Reviewer #3: No

2. Has the statistical analysis been performed appropriately and rigorously? 

Reviewer #1: Yes

Reviewer #2: Yes

Reviewer #3: No

3. Have the authors made all data underlying the findings in their manuscript fully available?

Reviewer #1: Yes

Reviewer #2: Yes

Reviewer #3: No

4. Is the manuscript presented in an intelligible fashion and written in standard English?

Reviewer #1: Yes

Reviewer #2: Yes

Reviewer #3: Yes

5. Review Comments to the Author

Reviewer #1: Important note: This review pertains only to ‘statistical aspects’ of the study and so ‘clinical aspects’ [like medical importance, relevance of the study, ‘clinical significance and implication(s)’ of the whole study, etc.] are to be evaluated [should be assessed] separately/independently. Further please note that any ‘statistical review’ is generally done under the assumption that (such) study specific methodological [as well as execution] issues are perfectly taken care of by the investigator(s). This review is not an exception to that and so does not cover clinical aspects {however, seldom comments are made only if those issues are intimately / scientifically related & intermingle with ‘statistical aspects’ of the study}.

COMMENTS:

I believe that the value of ‘r’ reported in line 73 is Pearson’s Correlation coefficient [(r=0.49; p<0.01)]. In the light of the following note on Assessing Pearson’s Correlation coefficient*, conclusion given in line 76 [that ‘The STST may not be used as a complete alternative to the 6MWT’] is highly appreciated.

* Statistical test usually used to assess significance of Pearson’s ‘Correlation coefficient (r)’ is ‘t’ [where t = { r � [(n-2) / (1-r2)] }for df=n-2, n is sample size] and here Ho is that the population/standard value of ‘r’ is zero. You need r=0.878 to be significant at 5% but you need r=0.273 if n=50 & you need only r=0.088 if n=500. ‘P-value’ heavily depends on sample size. Therefore, it is customary to conclude/interpret as a ‘Very strong positive correlation/association/relationship only if r = 0.70 or higher.

As you can see the ‘r’ [(r=0.49; p<0.01)] value here indicates a positive association/correlation/relationship which is not very strong though highly ‘significant’ statistically.

Further note that Pearson’s Correlation coefficient is not a measure of ‘Agreement/Concordance’ [instead ‘Kappa Coefficient’ (for categorical data) or ‘intra-class correlation coefficient ICC’ (for numerical/continuous data) are generally used for assessing/measuring Agreement/Concordance].

All these things are mentioned here because [according to lines 165-67: The sample size estimation showed that 36 participants were necessary to detect a correlation coefficient of 0.45 between the STST and 6MWT performances, with 80% power and an alpha level of 0.05] sample size estimation is done on the basis of detection of a correlation coefficient. Intra class correlation coefficient (ICC) were used to verify reliability [lines 182-84: Intra class correlation coefficient (ICC) were calculated using a random effects model for coherence of mean measures to verify reliability between the first and the second test for both STST and 6MWT] and not for assessing/measuring Agreement/Concordance between STST and 6MWT. Nevertheless, what is done is perfectly alright since Intra class correlation coefficient (ICC) is also a good measure of ‘reliability’.

A minor suggestion: In Table 2 [Table 2. Pre- and Post-exercise test cardio-respiratory response comparisons for the STST and 6MWT in children with CF] good that both ‘Pre- and Post-exercise’ values are given (essential as description), however comparison could have been done on ‘change scores’. I think that would have been more appropriate. Is not that?

I do not agree with account given in lines 307-318 that a strong correlation was found between STST and 6MWT in the 14 youngest children (≤10 years) just because r=0.60 {for above reasons}. I do not think that ‘this correlation may support the use of the STST as a first approach to functional exercise capacity assessment in children ≤10 years’. However, [as pointed out in ‘Important note’ in the beginning that] this review pertains only to ‘statistical aspects’ of the study and so ‘clinical aspects’ should be assessed separately.

Further please note that any ‘statistical review’ is generally done under the assumption that (such) study specific methodological [as well as execution] issues are perfectly taken care of by the investigator(s). This review is not an exception.

Reviewer #2: The study of XX and colleagues reports the sit-to-stand test as a valid and reliable tool to assess exercise capacity in children and adolescents with cystic fibrosis. Yields strengthen the assessment of STST with adequate measurement properties to assess children and adolescents with CF when 6MWT is not possible. Albeit methodology is adequate and well written, some aspects in the text remains to be confirmed.

There are some points that need to be clarified:

Major comments:

1. Although the aim of the study was to evaluate STST validity in children and adolescents with CF, other measures of measurement properties were performed too. To assure reader’s understanding specify the type of measurement property performed in the objective and throughout the manuscript.

2. Although sample size estimation was performed, the small sample size is one of the major issues of this study. Details about sample size calculation needs to be confirmed to assure reliability in results achieved. This calculation was performed based on a published study with CF or based on a pilot study? Please include this information and how was performed the sample size estimation.

3. Why was the second performance in STST and 6MWT used for correlation analysis? Studies and guidelines recommend using the best attempt from both tests. Authors need to review or justify this aspect.

4. Five assessors performed the assessments of STST and 6MWT in three different centers. As the assessor can influence the performance and results in the test, please specify more detailed information about the assessors and their training in the manuscript.

5. Line 222 and 223, page 12: “… results indicated that 6MWT produced a greater cardiorespiratory demand than the STST…”, indeed increase in HR and RR on 6MWT suggest this conclusion. Although, 6MWT have more time duration than STST. As well the function required in these filed tests are different. Why STST and 6MWT were used to assess the same aspect in functionality? Levy et al. (1994) demonstrated different domains in physical capacity. Authors must discuss time duration, function performance and type of domain in physical capacity required in these.

Minor comments:

1. In Line 98, Page 6: The population which was observed associations of STST with other clinical outcomes need to be specified.

2. Although study procedures of STST and 6MWT was well described, more details about days, order and time of rest in other clinical outcomes performed need to be included.

3. In Line 160, Page 9: “Performance during the two rounds of both STST and the 6MWT were recorded” is confusing. Please review this phrase.

4. In statistical section, please include the information and reference used to classify intraclass correlation coefficient.

5. Only the intra-rater analysis was performed through intraclass correlation coefficient. This information must be described in statistical section and specified in the results.

6. In table 1: Please specify the measure described in range. In the statistical section the authors described the data as interquartile range. Although this information is not clear in the table.

7. Table 1 and 2: Please describe the sample size with n in lower-case letter.

8. The results showed a better correlation of STST with 6MWT in children <10 years of age. Although it is a relevant analysis, the stratification of the sample was not described in the methods and the reason for carrying out the analysis with only part of the sample was not justified. Authors must review this point.

9. Line 267, page 15: Does the term physically active refers to patients who underwent to pulmonary rehabilitation? Although pulmonary rehabilitation improve clinical outcomes, does not appears to improve physical activity. Authors should review this part of the discussion.

10. Line 309, page 17: The authors described as strong correlation was found between STST and 6MWT. Please specify the reference to classify the results of the correlation.

11. Opposite to the findings of literature in other chronic respiratory diseases and the association of sit-to-stand movement with muscle activation/strength, no correlation was found between STST and quadriceps strength. The authors should discuss this finding in the article.

Reviewer #3: This is a study about the validity of sit to stand test in children with CF.

It was a Multicenter study in 3 French CF Centers

There are some issues that a should explored by the authors , described bellow.

Introduction

It is well described

Method

How many patients there is in the 3 French CF Centers? About that total number, how many were selected to this study?

It is necessary to describe, or to add reference, about the "clinically stable participants".

There is no reference for the STST. It should be there.

It is necessary to describe the outcomes, fot STST and 6MWT.

Heart rate, SpO2 and Borg were not discribed in the testes.

Why respiratory muscles were assessed? It is not included in the study aim.

The second test was used to evaluate the validity. Why not the best test?

Sample size

Even thought the sample was based on the expected correlation, it is known the validation study has to be, at least, 50 volunteers. The strength of quadriceps should be included as sample size estimation.

CPET was assessed? It was not included in the Method, only in the Statistical section.

Statistical Analysis

The Bland Altman plot should be addressed.

Results

Quality of life, Borg, blood pressure, respiratory rate (...) were not describe in the Method Section.

Table 1 is usually used to describe the population characteristics. The outcomes should be in table 2.

The was follow up? Why it is written tin the flow diagram?

Discussion and Conclusion

In my believe, it is not possible to conclude that STST is not valid test! The sample size is not enough to guarantee it.

STST has different information than 6MWT, specially for the test time. Thus, the authors should not expected to change on for other.

6. PLOS authors have the option to publish the peer review history of their article (what does this mean?). If published, this will include your full peer review and any attached files.

Reviewer #1: No

Reviewer #2: No

Reviewer #3: No

---

## [Author Response · Author response to Decision Letter 0]

16 Oct 2020

From: Yann Combret, October 2020

To: Dr. Andrea Martinuzzi

Academic Editor, Plos One

Dear Dr. Martinuzzi, 

Thank you for your invitation to provide a revised version of our work. We are sincerely grateful to you and the reviewers for the time you spent reviewing our paper entitled “Measurement properties of the one-minute sit-to-stand test in children and adolescents with cystic fibrosis: a multicenter randomized cross-over trial”. The comments made by the reviewers have been addressed in this revised version of our manuscript. The following is a point-by-point response to the reviewers’ comments. We have provided a marked-up version of our manuscript and a clean version along with this response according to your recommendations. This response letter is also provided as a supplementary file. We hope that you will find the responses and revisions satisfactory and that the manuscript now meets Plos One publication criteria.

 We look forward to hearing from you in the near future.

 Yours sincerely.

Yann COMBRET and Gregory REYCHLER

Please find below our point-by-point response to the reviewers (with modifications highlighted).

 

Response to Reviewer #1

I believe that the value of ‘r’ reported in line 73 is Pearson’s Correlation coefficient [(r=0.49; p<0.01)]. In the light of the following note on Assessing Pearson’s Correlation coefficient*, conclusion given in line 76 [that ‘The STST may not be used as a complete alternative to the 6MWT’] is highly appreciated. 

* Statistical test usually used to assess significance of Pearson’s ‘Correlation coefficient (r)’ is ‘t’ [where t = { r � [(n-2) / (1-r2)] }for df=n-2, n is sample size] and here Ho is that the population/standard value of ‘r’ is zero. You need r=0.878 to be significant at 5% but you need r=0.273 if n=50 & you need only r=0.088 if n=500. ‘P-value’ heavily depends on sample size. Therefore, it is customary to conclude/interpret as a ‘Very strong positive correlation/association/relationship only if r = 0.70 or higher.

Thank you for this comment. Indeed, the “r” refers to the Pearson’s correlation coefficient. We have slightly modified the interpretation of the correlation coefficient referring to criterion validity according to the clinical interpretations proposed by the two other reviewers. Nonetheless, the conclusion is still driven by the fact that the correlation is moderate. 

As you can see the ‘r’ [(r=0.49; p<0.01)] value here indicates a positive association/correlation/relationship which is not very strong though highly ‘significant’ statistically. Further note that Pearson’s Correlation coefficient is not a measure of ‘Agreement/Concordance’ [instead ‘Kappa Coefficient’ (for categorical data) or ‘intra-class correlation coefficient ICC’ (for numerical/continuous data) are generally used for assessing/measuring Agreement/Concordance]. All these things are mentioned here because [according to lines 165-67: The sample size estimation showed that 36 participants were necessary to detect a correlation coefficient of 0.45 between the STST and 6MWT performances, with 80% power and an alpha level of 0.05] sample size estimation is done on the basis of detection of a correlation coefficient. Intra class correlation coefficient (ICC) were used to verify reliability [lines 182-84: Intra class correlation coefficient (ICC) were calculated using a random effects model for coherence of mean measures to verify reliability between the first and the second test for both STST and 6MWT] and not for assessing/measuring Agreement/Concordance between STST and 6MWT. Nevertheless, what is done is perfectly alright since Intra class correlation coefficient (ICC) is also a good measure of ‘reliability’.

Thank you for approving our method. 

A minor suggestion: In Table 2 [Table 2. Pre- and Post-exercise test cardio-respiratory response comparisons for the STST and 6MWT in children with CF] good that both ‘Pre- and Post-exercise’ values are given (essential as description), however comparison could have been done on ‘change scores’. I think that would have been more appropriate. Is not that?

Thank you for this suggestion. The ‘change scores’ of the cardio-respiratory outcomes during both tests are now presented in Table 3 (‘Table 2’ renamed) with comparisons. 

I do not agree with account given in lines 307-318 that a strong correlation was found between STST and 6MWT in the 14 youngest children (≤10 years) just because r=0.60 {for above reasons}. I do not think that ‘this correlation may support the use of the STST as a first approach to functional exercise capacity assessment in children ≤10 years. However, [as pointed out in ‘Important note’ in the beginning that] this review pertains only to ‘statistical aspects’ of the study and so ‘clinical aspects’ should be assessed separately.

The clinical interpretation proposed by the reviewer is true. The interpretation has been modified according to the comments provided by the three reviewers. The concerned sentences have been removed from the ‘Discussion’ and ‘Conclusion’ sections. The discussion now focuses on the differences in functionality investigated by both tests, as well as the different durations of these tests and the clinical implications of our findings. 

Moreover, Reviewers #2 and #3 pointed out that the correlations should have been addressed using the best attempts of both tests (rather than the second attempt), which altered the results and the sentence concerning children ≤10 years was deleted accordingly. 

P18, L285: Discussion

“This moderate correlation can be explained by the different domains of physical capacity investigated by both tests, and the corresponding differences in energy expenditure [22,23]. The sit to stand movement is relevant for activities of daily living and is repeated frequently throughout the day by both adults and children. The lack of ability of the one-minute STST to predict the distance covered during a 6MWT can be attributed to the functional differences between the tests, as well as differences in energy utilization between walking for 6-minutes and standing repeatedly from a chair for 1-minute. Interestingly, a longer variation of the one-minute STST (the 3-minute STST) induced a similar functional response to that of the 6MWT in patients with COPD, leading the authors to conclude that the tests were equivalent [24]. Nevertheless, it should be noted that the 3-minute STST induced a higher degree of leg fatigue than the 6MWT, reinforcing the suggestion that the tests have different functional demands and should not be used interchangeably. The STST remains useful if the 6MWT cannot be performed.”

References

[22] Ridley K, Olds TS. Assigning Energy Costs to Activities in Children: A Review and Synthesis. Med Sci Sports Exerc. 2008;40: 1439–1446. doi:10.1249/MSS.0b013e31817279ef

[23] Butte NF, Watson KB, Ridley K, Zakeri IF, McMurray RG, Pfeiffer KA, et al. A Youth Compendium of Physical Activities: Activity Codes and Metabolic Intensities. Med Sci Sports Exerc. 2018;50: 246–256. doi:10.1249/MSS.0000000000001430

P20, L341: Discussion

“This difference in cardiorespiratory adaptation, which has previously been demonstrated in healthy adults performing the same tests, may be due to the different durations of each test [27]. Greater adaptation could be expected for a six-minute walk compared to a task like standing repeatedly from a chair for only one minute. Furthermore, walking could involve a larger number of muscles, inducing a faster HR and a higher level of leg fatigue and dyspnea. These differences further strengthen the hypothesis that these tests measure different aspects of functional capacity.”

P21, L362: Discussion

“Furthermore, it is important to note that the STST does not measure the same aspects of functional capacity as the 6MWT, and that its discriminative validity as well as its clinical applications (e.g. the association with hospitalizations for CF or other important prognostic outcomes) have yet to be demonstrated.”

P22, L377: Conclusion

“The results of this study suggest the STST measures different aspects of functional capacity than the 6MWT in children with CF. STST performance was moderately correlated with 6MWD, and the test induced a lower cardio-respiratory response. Neither quadriceps nor respiratory muscle strength were related to the number of STST repetitions, and the clinical value of the test is still to be proven. The simplicity of the STST for both children and practitioners makes it a reasonable second option when 6MWT or CPET cannot be performed, but healthcare providers should be aware that these tests cannot be used interchangeably.” 

Response to Reviewer #2

The study of XX and colleagues reports the sit-to-stand test as a valid and reliable tool to assess exercise capacity in children and adolescents with cystic fibrosis. Yields strengthen the assessment of STST with adequate measurement properties to assess children and adolescents with CF when 6MWT is not possible. Albeit methodology is adequate and well written, some aspects in the text remains to be confirmed.

Thank you for this careful revision. Please find below a point-by-point response to your queries. 

There are some points that need to be clarified:

Major comments:

1. Although the aim of the study was to evaluate STST validity in children and adolescents with CF, other measures of measurement properties were performed too. To assure reader’s understanding specify the type of measurement property performed in the objective and throughout the manuscript.

Thank you for this suggestion. The reviewer is right and we aimed to investigate STST1’ (one-minute sit-to-stand test) criterion validity to measure functional exercise capacity (compared to 6MWT results) but also intra-rater reliability and repeatability. The term ‘criterion validity’ (according to COSMIN guidelines) has been used rather than ‘validity’ to specify the property analyzed. The other measurement properties have been repeatedly specified throughout the manuscript. The following changes have been made throughout the manuscript. 

P1, L1: Title 

“Measurement properties of the one-minute sit-to-stand test in children and adolescents with cystic fibrosis: a multicenter randomized cross-over trial”

P5, L60: Abstract

“This study aimed to evaluate STST measurement properties in children and adolescents with CF.”

P5, L65: Abstract

“Intra-rater reliability, test-retest repeatability, mean bias and limits of agreement were also assessed.”

P5, L71: Abstract

“Both tests had very good intra-rater reliability (ICCSTST=0.91 (95%CI 0.76-0.96) and ICC6MWT=0.94 (95%CI 0.85-0.97)), and a significant test-retest learning effect.”

P7, L103: Introduction

“(1) explore STST criterion validity to measure functional exercise capacity compared to the 6MWT in children with CF; (2) evaluate STST intra-rater reliability, test-retest reliability, mean bias and limits of agreement; and (3) analyse correlations with other outcomes previously identified as accounting for functional exercise capacity (respiratory and quadriceps muscle strength, nutritional status, etc.).”

P17, L270: Discussion

“This study showed that the STST had moderate criterion validity for the measurement of functional exercise capacity when compared to the 6MWT (r=0.49) in the entire study group of children with CF and in children ≤10 years of age (r=0.54).”

P17, L274: Discussion

“A significant learning effect and a very good intra-rater reliability were reported for both tests.”

2. Although sample size estimation was performed, the small sample size is one of the major issues of this study. Details about sample size calculation needs to be confirmed to assure reliability in results achieved. This calculation was performed based on a published study with CF or based on a pilot study? Please include this information and how was performed the sample size estimation.

Thank you for this question. This point has also been raised by reviewer #3 and it appears clearly that this requires further explanation. Actually, the sample size estimation for this study based on correlation analysis (i.e. criterion validity between two tests) was based on the estimation of the correlation coefficient ‘r’, α (=0.05) and β (usually set at 20% to ensure an 80% power). However, in that particular case, the correlation coefficient estimation can be tricky because a high ‘r’ will lead to a very limited sample size. Two previous studies were initially identified to estimate ‘r’ and estimate the necessary sample size for the present study. The first one (Gurses HN, Zeren M, Denizoglu Kulli H, Durgut E. The relationship of sit-to-stand tests with 6-minute walk test in healthy young adults: Medicine (Baltimore). 2018;97: e9489. doi:10.1097/MD.0000000000009489) reported a correlation coefficient of r=0.647 between STST1’ and 6MWT in healthy young adults. The second one (Radtke T, Puhan MA, Hebestreit H, Kriemler S. The 1-min sit-to-stand test—A simple functional capacity test in cystic fibrosis? J Cyst Fibros. 2016;15: 223–226. doi: 10.1016/j.jcf.2015.08.006) revealed a r=0.627 between STST1’ and VO2peak in adults with CF. 

A sample size estimation based on one of these studies would have been: 

- r=0.647, α=0.05 and β=0.20 → n=16 participants (or 21 participants for a 90% power)

- r=0.627, α=0.05 and β=0.20 → n=17 participants (or 22 participants for a 90% power)

According to your comment, this would have been too small a sample size for the purpose of the present study. Therefore, we aimed to perform a sample size estimation to be adequately powered to detect ‘at least’ a moderate correlation between the STST and the 6MWT. In previous publications, a moderate correlation is defined by a ‘r’ of 0.40 or 0.50 (according to Swinscow TDV, Campbell MJ. Statistics at square one. 10., [upgraded] ed. London: BMJ; 2002. 158 p or Hinkle DE, Wiersma W, Jurs SG (2003). Applied Statistics for the Behavioral Sciences 5th ed. Boston: Houghton Mifflin) which is why we aimed to compute a sample size estimation using a moderate ‘r’ value of 0.45 according to previous publications on the same topic (Reychler G, Boucard E, Peran L, Pichon R, Le Ber-Moy C, Ouksel H, Liistro G, Chambellan A, Beaumont M. One-minute sit-to-stand test is an alternative to 6MWT to measure functional exercise performance in COPD patients. Clin Respir J. 2018; 12(3): 1247-1256. doi: 10.1111/crj.12658). 

The following changes have been made: 

P11, L188: Methods

“The sample size estimation showed that 36 participants were necessary to detect at least a moderate correlation with a correlation coefficient of 0.45 between the STST and 6MWT performances, with 80% power and an alpha level of 0.05 [17,18].”

[17] Hinkle D, Wiersma W, Jurs S. Applied Statistics for the Behavioral Sciences. 5th ed. Boston: Houghton Mifflin; 2003. 

[18] Swinscow TDV, Campbell MJ. Statistics at square one. 10., [upgraded] ed. London: BMJ; 2002.

3. Why was the second performance in STST and 6MWT used for correlation analysis? Studies and guidelines recommend using the best attempt from both tests. Authors need to review or justify this aspect.

Thank you for this pertinent correction. Indeed, the correlation analysis, as well as the comparisons of the cardio-respiratory responses during both tests, should have been undertaken using the best attempt of both tests rather than the second iterations. This has been corrected in the methods and results section, and throughout the manuscript. 

The new analyses undertaken revealed that the stronger correlation reported in the previous version for younger children (r=0.60) was no longer found when the best attempt of each test was considered. The conclusions on that point have been adapted according to these new results. Similarly, the STST was no longer correlated with MEP when the best attempt of the test was considered and the conclusions have been modified accordingly. 

P5, L69: Abstract

“On average, 39.6 ±10.5 repetitions were performed during the STST and mean 6MWD was 596.0 ±102.6 meters.”

P5, L70: Abstract

“STST number of repetitions was significantly correlated with 6MWD (r=0.48; p<0.01).”

P5, L76: Abstract

“The STST is an easy-to-use functional test with moderate criterion validity when compared to the 6MWT in children with CF, probably because both tests measure different components of functional exercise capacity. The STST is useful when the 6MWT is unfeasible, however further investigations are required to explore the clinical implications of STST results in children with CF.”

P11, L179: Methods

“The best performances of each test were used to evaluate criterion validity and correlations with other outcomes.”

P11, L183: Methods

“Cardio-respiratory adaptations during the best performance of each test were also compared.”

P14, L225: Results

“The average number of repetitions performed in the second STST was 39.6 ±10.5 and the distance walked in the second 6MWT was 596.0 ±102.6 metres (Table 2).”

P15, L234: Results

“The number of STST repetitions was correlated with 6MWD (r=0.48; p<0.01) in the entire sample of children. Criterion validity of the STST with the 6MWT was equivalent in children ≤ 10 years of age (r=0.54; p<0.05) (Fig 2).”

P15, L239: Results

“The number of STST repetitions was neither correlated to QS (r=-0.01; p=0.99), MIP (r=0.31; p=0.07), MEP (r=0.32, p=0.06) nor lung function (S1 Table). STST was not correlated with HRQOL items, except for child-estimated social functioning (r=0.37; p=0.03) (S2 Table). Inversely, 6MWD correlated with every muscle strength assessment (r=0.48; r=0.54 and r=0.46; p<0.01 for MIP, MEP and QS respectively) and with lung function (r=0.59; r=0.55 and r=0.58; p<0.01 for FVC, FEV1 and PEF respectively) (S1 Table). 6MWD was associated with several HRQOL items including child physical functioning (S2 Table).”

P17, L254: Results

“Changes in HR during both tests correlated with participant height (r=0.38 and r=0.42; p=0.02 for STST and 6MWT respectively). Delta HR during the STST correlated with lung function (r=0.36, r=0.46 and r=0.50; p<0.05 with FVC, FEV1 and PEF respectively) and with child-reported HRQOL total scores (r=0.46; p<0.01) whereas delta HR during 6MWT correlated with FEV1 and PEF (r=0.35 and r=0.51; p=0.04 and p<0.01 respectively).”

The results of the best test have been added to the appropriate section in Table 1. The correlation coefficients and the p-values associated have also been added in supplemental Tables 1 and 2. Furthermore, the results of the cardio-respiratory adaptations in Table 3 have been replaced by the values achieved during the best attempt of each test. 

P17, L270: Discussion

“This study showed that the STST had moderate criterion validity for the measurement of functional exercise capacity when compared to the 6MWT (r=0.49) in the entire study group of children with CF and in children ≤10 years of age (r=0.54). The STST was not correlated to other outcomes whereas the 6MWT was correlated with muscle strength (respiratory muscles and quadriceps) and HRQOL.”

P19, L311: Discussion

“The mean number of STST repetitions performed in the present study was around 40 (±10).”

P19, L323: Discussion

“Surprisingly, the average 6MWD of 596m in this study was considerably lower than the average distance of 742m reported previously in a sample of children of similar age and lung function [6].”

P20, L335: Discussion

“End-exercise HR during the 6MWT reported in studies including healthy children and children with CF were considerably higher than in the present study with respectively 158 and 154 beats/min compared to 126 beats/min [33, 37].”

P20, L337: Discussion

“Another recent study of healthy children reported a higher peak HR during the STST compared to our study (139 compared to 116 beats/min) [11].”

4. Five assessors performed the assessments of STST and 6MWT in three different centers. As the assessor can influence the performance and results in the test, please specify more detailed information about the assessors and their training in the manuscript.

Thank you for this question. All the assessors were trained by the principal investigator (YC) of the present study. A visit was performed in the Rouen and Caen centers prior to the first inclusion to train the investigators for both functional tests and for respiratory and quadriceps strength testing as well as HRQoL rating. All the investigators are senior physiotherapists trained in muscle and functional exercise capacities testing with at least 5 years of experience in that domain. Furthermore, the procedure for the 6MWT was extracted from the well-described ‘technical standards’ of the ERS/ATS guidelines and the standardized sentences were taught to all the investigators during this preliminary visit. Similarly, standardized sentences were used for the STST and the procedure was based on the protocol previously described by Radtke et al. (Radtke T, Puhan MA, Hebestreit H, Kriemler S. The 1-min sit-to-stand test—A simple functional capacity test in cystic fibrosis? J Cyst Fibros. 2016;15: 223–226. doi: 10.1016/j.jcf.2015.08.006). To minimize the effect of the assessor during both tests, no encouragements were allowed in any of the centers. 

The following sentences have been added: 

P8, L124: Methods

“All the investigators were senior physiotherapists, experienced (at least 5 years) in the assessment of muscle and functional capacities in individuals with CF. Furthermore, a preliminary visit was made to the Rouen and Caen centers by the principal investigator (YC) prior to the first inclusion to homogenize testing procedures.”

5. Line 222 and 223, page 12: “… results indicated that 6MWT produced a greater cardiorespiratory demand than the STST…”, indeed increase in HR and RR on 6MWT suggest this conclusion. Although, 6MWT have more time duration than STST. As well the function required in these filed tests are different. Why STST and 6MWT were used to assess the same aspect in functionality? Levy et al. (1994) demonstrated different domains in physical capacity. Authors must discuss time duration, function performance and type of domain in physical capacity required in these.

Thank you for this comment. Indeed, the reviewer is right and the differences in cardiorespiratory responses are probably strongly related to the duration of both tests, and the different aspects of functionality measured. This point is now emphasized and has been discussed several times in the Discussion section to strengthen the interpretation of our findings. The reviewer will notice that we are not referring to the aforementioned study by Levy et al. (1994) because we were unable to find this study with this information. However, the different aspects of functionality have been justified with the differences in energy expenditure referring to the Youth Compendium of Physical Activities. Of course, we would be pleased to include the reference proposed by the reviewer if you provide us more details. 

P18, L285: Discussion

“This moderate correlation can be explained by the different domains of physical capacity investigated by both tests, and the corresponding differences in energy expenditure [22,23]. The sit to stand movement is relevant for activities of daily living and is repeated frequently throughout the day by both adults and children. The lack of ability of the one-minute STST to predict the distance covered during a 6MWT can be attributed to the functional differences between the tests, as well as differences in energy utilization between walking for 6-minutes and standing repeatedly from a chair for 1-minute. Interestingly, a longer variation of the one-minute STST (the 3-minute STST) induced a similar functional response to that of the 6MWT in patients with COPD, leading the authors to conclude that the tests were equivalent [24]. Nevertheless, it should be noted that the 3-minute STST induced a higher degree of leg fatigue than the 6MWT, reinforcing the suggestion that the tests have different functional demands and should not be used interchangeably. The STST remains useful if the 6MWT cannot be performed.”

References

[22] Ridley K, Olds TS. Assigning Energy Costs to Activities in Children: A Review and Synthesis. Med Sci Sports Exerc. 2008;40: 1439–1446. doi:10.1249/MSS.0b013e31817279ef

[23] Butte NF, Watson KB, Ridley K, Zakeri IF, McMurray RG, Pfeiffer KA, et al. A Youth Compendium of Physical Activities: Activity Codes and Metabolic Intensities. Med Sci Sports Exerc. 2018;50: 246–256. doi:10.1249/MSS.0000000000001430

P18, L298: Discussion

“In contrast, the STST has previously been shown to provide an adequate reflection of anaerobic capacities and consequently of quadriceps strength, even though there are conflicting results in the literature [10,25–27]. In the present study, quadriceps strength was not correlated with the number of STST repetitions. We suggest three reasons for this lack of correlation. Firstly, other factors, such as balance or quadriceps endurance (two components not measured in this study), also contribute to the sit-to-stand movement, thus quadriceps strength alone cannot fully explain STST performance [28,29]. Secondly, although quadriceps strength testing using fixed handheld dynamometry has been shown to be reliable and accurate for patients with chronic respiratory insufficiency, its reliability in children with CF has been little studied [30,31]. Finally, the results for both quadriceps strength and STST performance may have been affected by the age and height of the children in this study. This effect could have been reduced by normalising the results by leg length: evidence suggests that the product of STST performance and body weight produce higher correlations with lower limb strength [10,26].”

References

[25] Reychler G, Boucard E, Peran L, Pichon R, Le Ber-Moy C, Ouksel H, et al. One minute sit-to-stand test is an alternative to 6MWT to measure functional exercise performance in COPD patients. Clin Respir J. 2018;12: 1247–1256. doi:10.1111/crj.12658

[26] Reychler G, Pincin L, Audag N, Poncin W, Caty G. One-minute sit-to-stand test as an alternative tool to assess the quadriceps muscle strength in children. Respir Med Res. 2020;78: 100777. doi: 10.1016/j.resmer.2020.100777

[27] Ozalevli S, Ozden A, Itil O, Akkoclu A. Comparison of the Sit-to-Stand Test with 6min walk test in patients with chronic obstructive pulmonary disease. Respir Med. 2007;101: 286–293. doi: 10.1016/j.rmed.2006.05.007

[28] Gruet M, Decorte N, Mely L, Vallier J-M, Camara B, Quetant S, et al. Skeletal muscle contractility and fatigability in adults with cystic fibrosis. J Cyst Fibros. 2016;15: e1–e8. doi: 10.1016/j.jcf.2015.05.004

[29] Lima TRL, Guimarães FS, Sá Ferreira A, Penafortes JTS, Almeida VP, Lopes AJ. Correlation between posture, balance control, and peripheral muscle function in adults with cystic fibrosis. Physiother Theory Pract. 2014;30: 79–84. doi:10.3109/09593985.2013.820246

[30] Bui K-L, Mathur S, Dechman G, Maltais F, Camp P, Saey D. Fixed Handheld Dynamometry Provides Reliable and Valid Values for Quadriceps Isometric Strength in People With Chronic Obstructive Pulmonary Disease: A Multicenter Study. Phys Ther. 2019;99: 1255–1267. doi:10.1093/ptj/pzz059

[31] Combret Y, Medrinal C, Bonnevie T, Gravier F-E, Le Roux P, Lamia B, et al. Clinimetric evaluation of muscle function tests for individuals with cystic fibrosis: A systematic review. J Cyst Fibros. 2020; S1569199320301648. doi: 10.1016/j.jcf.2020.05.014

P20, L341: Discussion

“This difference in cardiorespiratory adaptation, which has previously been demonstrated in healthy adults performing the same tests, may be due to the different durations of each test [27]. Greater adaptation could be expected for a six-minute walk compared to a task like standing repeatedly from a chair for only one minute. Furthermore, walking could involve a larger number of muscles, inducing a faster HR and a higher level of leg fatigue and dyspnea. These differences further strengthen the hypothesis that these tests measure different aspects of functional capacity.”

P21, L362: Discussion

“Furthermore, it is important to note that the STST does not measure the same aspects of functional capacity as the 6MWT, and that its discriminative validity as well as its clinical applications (e.g. the association with hospitalizations for CF or other important prognostic outcomes) have yet to be demonstrated.”

P22, L377: Conclusion

“The results of this study suggest the STST measures different aspects of functional capacity than the 6MWT in children with CF. STST performance was moderately correlated with 6MWD, and the test induced a lower cardio-respiratory response. Neither quadriceps nor respiratory muscle strength were related to the number of STST repetitions, and the clinical value of the test is still to be proven. The simplicity of the STST for both children and practitioners makes it a reasonable second option when 6MWT or CPET cannot be performed, but healthcare providers should be aware that these tests cannot be used interchangeably.”

Minor comments:

1. In Line 98, Page 6: The population which was observed associations of STST with other clinical outcomes need to be specified.

The reviewer is right. Both studies cited to justify this sentence have been conducted with adults with CF. This has now been specified. 

P7, L96: Introduction

“Derived STST measures (PowerSTST or STST x bodyweight=STSTW) have also shown strong associations with VO2peak, maximal power and quadriceps muscle strength in adults with CF [9,10].”

2. Although study procedures of STST and 6MWT was well described, more details about days, order and time of rest in other clinical outcomes performed need to be included.

Thank you for this suggestion. All the measures were undertaken within the same day. On arrival, anthropometrics characteristics of each participant were noted, included lung function if the last spirometry was more than 6-months old. If not, lung function measurement was undertaken according to international guidelines. Then, respiratory and quadriceps muscle testing was undertaken. Each measure of MIP, MEP and quadriceps MVC was separated by a 60 second rest period. Functional testing was then undertaken according to the protocol previously described. Finally, the HRQoL questionnaire was rated by the participant (and his/her parents for the youngest children) during the rest period between two functional tests. The following changes have been made in the Methods section to further describe the testing protocol.

P10, L155: Methods

“Participant characteristics were recorded upon arrival, including lung function if the last spirometry was more than 6-months old. Otherwise, lung function was evaluated according to ATS/ERS guidelines prior to the muscle and functional testing [13]. Then, before the functional testing, respiratory muscle strength was assessed using maximal inspiratory and expiratory pressures (MIP and MEP), measured with an electronic manometer (MicroRPM, Eolys).”

P10, L161: Methods

“. The best of three inspiratory and expiratory manoeuvres that varied by less than 10%, with a 60 second rest period between each measurement, were recorded and used for analysis [14]. Moreover, quadriceps strength (QS) was also assessed prior to functional testing using an isometric measurement of the maximal voluntary contraction of the dominant limb with a hand-held dynamometer (Microfet2, Biometrics).”

P10, L170: Methods

“After two repetitions for familiarization, 3 trials were conducted, with a 60 second rest between each measurement, and the best used in the analysis.”

P11, L174: Methods

“Functional testing was then undertaken according to the protocol described above and health-related quality of life (HRQOL) was evaluated using the French version of the CFQ-R during the rest period between two iterations of the field tests [16].”

3. In Line 160, Page 9: “Performance during the two rounds of both STST and the 6MWT were recorded” is confusing. Please review this phrase.

Thank you for this suggestion. The sentence has been modified to describe which ‘performance’ was measured during the two iterations of both functional tests. 

P11, L178: Methods

“The number of repetitions achieved during the two STST and the distance achieved during the two 6MWT were recorded.”

4. In statistical section, please include the information and reference used to classify intraclass correlation coefficient.

Thank you for this suggestion. The classification used was the one from Portney and Watkins that proposed the following interpretation for ICC values: <0.5 = poor reliability; 0.5-0.75 = moderate reliability; 0.75-0.90 = good reliability; >0.90 = very good or excellent reliability. The classification as well as the reference have been added to the concerned section. 

P12, L210: Methods

“ICC values were interpretated as follows: <0.5 = poor reliability; 0.5-0.75 = moderate reliability; 0.75-0.90 = good reliability; >0.90 = very good reliability [19].”

[19] Portney L, Watkins M. Foundations of clinical research: applications to practice. New Jersey: Prentice Hall; 2000. 

5. Only the intra-rater analysis was performed through intraclass correlation coefficient. This information must be described in statistical section and specified in the results.

Thank you for this pertinent comment. The word ‘intra-rater’ has been added to the relevant sections to further precise the analysis undertaken. 

P12, L206: Methods

“Intra class correlation coefficients (ICC) were calculated using a random effects model for coherence of mean measures to verify intra-rater reliability between the first and the second test for both STST and 6MWT.”

P17, L262: Results

“Both tests showed very good intra-rater reliability with both ICCs above 0.9 (ICCSTST=0.91 (95%CI 0.76-0.96) and ICC6MWT=0.94 (95%CI 0.85-0.97)).”

6. In table 1: Please specify the measure described in range. In the statistical section the authors described the data as interquartile range. Although this information is not clear in the table.

Thank you for this question. The ‘range’ values in the Table 1 represent the complete range of values, from minimum to maximum, reached by the participants. The ‘interquartile range’ has been used for the sole description of CF exacerbations and hospitalizations in the Table. This has been specified in the Table 1 legend. 

P12, L195: Methods

“Data are expressed as numbers and percentages for categorical variables and as means (±standard deviations) or medians (interquartile range) for quantitative variables and the range of participants’ values (minimum-maximum) for the sample characteristics.”

P13, L221: Table 1

“Data are shown as either means (±standard deviations) or medians (interquartile ranges), and ranges (minimum-maximum).” 

7. Table 1 and 2: Please describe the sample size with n in lower-case letter.

Thank you. This has been corrected in Tables 1 and 2. 

8. The results showed a better correlation of STST with 6MWT in children <10 years of age. Although it is a relevant analysis, the stratification of the sample was not described in the methods and the reason for carrying out the analysis with only part of the sample was not justified. Authors must review this point.

Thank you for this pertinent comment. This stratification was designed due to the latest international statement on exercise testing for children with CF (Hebestreit H, Arets HGM, Aurora P, Boas S, Cerny F, Hulzebos EHJ, et al. Statement on Exercise Testing in Cystic Fibrosis. Respiration. 2015;90: 332–351. doi:10.1159/000439057) that acknowledges that exercise testing should be performed in children older than 10 using either CPET or 6MWT according to available resources. The main aim of this secondary analysis was to provide useful results for clinicians for both young children and adolescents. This has been clarified in the methods section. 

P11, L180: Methods

“According to the latest international statement, exercise testing (using either CPET or 6MWT) should be undertaken yearly in all individuals with CF from the age of 10 [2]. We therefore also aimed to investigate STST criterion validity separately in the children ≤ 10 years of age and those aged 11 – 18 years. Cardio-respiratory adaptations during the best performance of each test were also compared.”

P12, L199: Methods

“Criterion validity and relationships with other outcomes were explored using Spearman or Pearson correlation coefficients in the complete study sample, in children ≤10 years of age and in children aged 11-18 years.” 

9. Line 267, page 15: Does the term physically active refers to patients who underwent to pulmonary rehabilitation? Although pulmonary rehabilitation improve clinical outcomes, does not appears to improve physical activity. Authors should review this part of the discussion.

Thank you for this very interesting suggestion. The reviewer is right and the term ‘physically active’ was misleading. Actually, the 67 repetitions performed by the participants in the study by Radtke et al. were measured at the end of a 3-week rehabilitation program. The aim of this sentence was therefore to emphasize that these individuals would probably have a higher functional exercise capacity at the end of this rehabilitation program that patients who would not take part in such program. The sentence has been modified to better describe what we mean. 

P19, L313: Discussion

“Adults with CF at the end of a 3-week pulmonary rehabilitation program performed 67 repetitions in a previous study but it would be expected that even with CF, at the end of a rehabilitation program adults would have higher exercise capacities than children [8].”

10. Line 309, page 17: The authors described as strong correlation was found between STST and 6MWT. Please specify the reference to classify the results of the correlation.

Thank you for this suggestion. The reference used was the one by Swinscow and Campbell previously cited to answer the reviewer (and added as [18]). The concerned sentence has been modified since the correlation reported for young children was slightly lower in the new analysis undertaken with the best attempt of each functional test (rather than the second attempt). Furthermore, the classification has been added to the statistical analysis section for clarity. 

P12, L202: Methods

“Correlation coefficients were interpretated as follows: 0 to 0.19 = very weak correlation, 0.20 to 0.39 = weak correlation, 0.40 to 0.59 = moderate correlation, 0.60 to 0.79 = strong correlation and 0.80 to 1.0 = very strong correlation [18].”

11. Opposite to the findings of literature in other chronic respiratory diseases and the association of sit-to-stand movement with muscle activation/strength, no correlation was found between STST and quadriceps strength. The authors should discuss this finding in the article.

Thank you for this comment. Quadriceps strength has indeed been linked with STST performance in previous studies, even though results in CF patients are conflicting. This point has been specified and discussed. Several hypotheses have been raised:

P18, L298: Discussion

“In contrast, the STST has previously been shown to provide an adequate reflection of anaerobic capacities and consequently of quadriceps strength, even though there are conflicting results in the literature [10,25–27]. In the present study, quadriceps strength was not correlated with the number of STST repetitions. We suggest three reasons for this lack of correlation. Firstly, other factors, such as balance or quadriceps endurance (two components not measured in this study), also contribute to the sit-to-stand movement, thus quadriceps strength alone cannot fully explain STST performance [28,29]. Secondly, although quadriceps strength testing using fixed handheld dynamometry has been shown to be reliable and accurate for patients with chronic respiratory insufficiency, its reliability in children with CF has been little studied [30,31]. Finally, the results for both quadriceps strength and STST performance may have been affected by the age and height of the children in this study. This effect could have been reduced by normalising the results by leg length: evidence suggests that the product of STST performance and body weight produce higher correlations with lower limb strength [10,26].”

References

[25] Reychler G, Boucard E, Peran L, Pichon R, Le Ber-Moy C, Ouksel H, et al. One-minute sit-to-stand test is an alternative to 6MWT to measure functional exercise performance in COPD patients. Clin Respir J. 2018;12: 1247–1256. doi:10.1111/crj.12658

[26] Reychler G, Pincin L, Audag N, Poncin W, Caty G. One-minute sit-to-stand test as an alternative tool to assess the quadriceps muscle strength in children. Respir Med Res. 2020;78: 100777. doi: 10.1016/j.resmer.2020.100777

[27] Ozalevli S, Ozden A, Itil O, Akkoclu A. Comparison of the Sit-to-Stand Test with 6min walk test in patients with chronic obstructive pulmonary disease. Respir Med. 2007;101: 286–293. doi: 10.1016/j.rmed.2006.05.007

[28] Gruet M, Decorte N, Mely L, Vallier J-M, Camara B, Quetant S, et al. Skeletal muscle contractility and fatigability in adults with cystic fibrosis. J Cyst Fibros. 2016;15: e1–e8. doi: 10.1016/j.jcf.2015.05.004

[29] Lima TRL, Guimarães FS, Sá Ferreira A, Penafortes JTS, Almeida VP, Lopes AJ. Correlation between posture, balance control, and peripheral muscle function in adults with cystic fibrosis. Physiother Theory Pract. 2014;30: 79–84. doi:10.3109/09593985.2013.820246

[30] Bui K-L, Mathur S, Dechman G, Maltais F, Camp P, Saey D. Fixed Handheld Dynamometry Provides Reliable and Valid Values for Quadriceps Isometric Strength in People With Chronic Obstructive Pulmonary Disease: A Multicenter Study. Phys Ther. 2019;99: 1255–1267. doi:10.1093/ptj/pzz059

[31] Combret Y, Medrinal C, Bonnevie T, Gravier F-E, Le Roux P, Lamia B, et al. Clinimetric evaluation of muscle function tests for individuals with cystic fibrosis: A systematic review. J Cyst Fibros. 2020; S1569199320301648. doi: 10.1016/j.jcf.2020.05.014

 

Response to Reviewer #3

Reviewer #3: This is a study about the validity of sit to stand test in children with CF.

It was a Multicenter study in 3 French CF Centers

There are some issues that a should explored by the authors, described below.

Thank you for this careful revision. Please find below a point-by-point response to your queries. 

Introduction

It is well described

Method

How many patients there is in the 3 French CF Centers? About that total number, how many were selected to this study?

Thank you for this question. Two of the CF Centers that included participants are large reference CF Centers (Rouen and Caen). Eighty CF children are followed in the Rouen CF Center and 70 in the Caen CF Center. Twenty-seven children are followed in the Le Havre CF Center. Among these children, 59 were aged 6-18 in Rouen at the time of the inclusions, 58 in Caen and 14 in Le Havre. The sample was constituted with the 49 first children approached and assessed for eligibility (see Figure 1 for the details) until 36 children were finally included according to the sample size estimation. 

It is necessary to describe, or to add reference, about the "clinically stable participants".

Thank you for this pertinent comment. ‘Clinically stable’ refers to the absence of hospitalization or exacerbations (i.e. intravenous or oral antibiotics use) within the last 4 weeks preceding inclusion. This has been specified in the concerned sentence. 

P8, L111: Methods

“In order to be included, they had to have a diagnosis of CF (sweat chloride > 60mmol/L), be aged between 6 and 18 years and to have been clinically stable (absence of hospitalizations or exacerbations requiring intravenous or oral antibiotics) for at least 4 weeks prior to the study.”

There is no reference for the STST. It should be there.

Thank you for this suggestion. The protocol for the STST procedure is now referenced using the protocol previously described by Strassman et al. 

P9, L135: Methods

“Participants were instructed to stand-up completely and sit-down at their own pace for 1 minute, without using their arms as leverage or support, as previously described [12].”

[12] Strassmann A, Steurer-Stey C, Lana KD, Zoller M, Turk AJ, Suter P, et al. Population-based reference values for the 1-min sit-to-stand test. Int J Public Health. 2013;58: 949–953. doi:10.1007/s00038-013-0504-z

It is necessary to describe the outcomes, for STST and 6MWT.

Thank you for this rightful suggestion. The outcomes have been described for both tests. 

P9, L140: Methods

“The number of completed STST cycles was recorded by the investigators.”

P9, L150: Methods

“The total distance covered (6MWD) was recorded by the investigators to the nearest meter.”

Heart rate, SpO2 and Borg were not described in the tests.

Thank you for this comment. The recording of cardio-respiratory adaptation has been mentioned and specified for each testing procedure. 

P9, L142: Methods

“Heart rate (HR), pulsed oxygen saturation (SpO2), systolic and diastolic blood pressure (SBP, DBP), and respiratory rate (RR) were measured during a 1-min sitting rest period before and after each STST (53NOP portable monitor, WelchAllyn). Dyspnea and lower limb fatigue were also recorded using the modified Borg scale.”

P10, L151: Methods

“HR, SpO2, SBP, DBP, RR, dyspnea and lower limb fatigue were also recorded during a 1-min sitting rest period before and after each 6MWT.”

Why respiratory muscles were assessed? It is not included in the study aim.

Thank you for this question. Respiratory muscle strength has previously been identified as a factor associated with peripheral muscle strength and exercise performance (see for instance Papalexopoulou et al. Respir Med, 2018 or Combret et al. J Cyst Fibros, 2020). Hence, we aimed to investigate the outcomes correlated with STST performance among the factors already known to contribute to functional exercise capacity (i.e. respiratory and peripheral muscle strength or nutritional status for instance). The following changes have been made in the study aims to emphasize this objective. 

P7, L102: Introduction

“Our study aimed to: (1) explore STST criterion validity to measure functional exercise capacity compared to the 6MWT in children with CF; (2) evaluate STST intra-rater reliability, test-retest reliability, mean bias and limits of agreement; and (3) analyse correlations with other outcomes previously identified as accounting for functional exercise capacity (respiratory and quadriceps muscle strength, nutritional status, etc.).”

The second test was used to evaluate the validity. Why not the best test?

Thank you for this pertinent question. This very important point was also raised by Reviewer#2. All the concerned analyses (i.e. validity, correlations with other outcomes, and cardio-respiratory adaptations comparisons) have been modified accordingly and are now based on the best performance achieved for each test. 

The list of changes undertaken because of this new analysis is detailed below. 

The new analyses undertaken revealed that the stronger correlation reported in the previous version for younger children (r=0.60) was no longer found when the best attempt of each test was considered. The conclusions on that point have been adapted according to these new results. Similarly, the STST was no longer correlated with MEP when the best attempt of the test was considered and the conclusions have been modified accordingly. 

P5, L69: Abstract

“On average, 39.6 ±10.5 repetitions were performed during the STST and mean 6MWD was 596.0 ±102.6 meters.”

P5, L70: Abstract

“STST number of repetitions was significantly correlated with 6MWD (r=0.48; p<0.01).”

P5, L76: Abstract

“The STST is an easy-to-use functional test with moderate criterion validity when compared to the 6MWT in children with CF, probably because both tests measure different components of functional exercise capacity. The STST is useful when the 6MWT is unfeasible, however further investigations are required to explore the clinical implications of STST results in children with CF.”

P11, L179: Methods

“The best performances of each test were used to evaluate criterion validity and correlations with other outcomes.”

P11, L183: Methods

“Cardio-respiratory adaptations during the best performance of each test were also compared.”

P14, L225: Results

“The average number of repetitions performed in the second STST was 39.6 ±10.5 and the distance walked in the second 6MWT was 596.0 ±102.6 metres (Table 2).”

P15, L234: Results

“The number of STST repetitions was correlated with 6MWD (r=0.48; p<0.01) in the entire sample of children. Criterion validity of the STST with the 6MWT was equivalent in children ≤ 10 years of age (r=0.54; p<0.05) (Fig 2).”

P15, L239: Results

“The number of STST repetitions was neither correlated to QS (r=-0.01; p=0.99), MIP (r=0.31; p=0.07), MEP (r=0.32, p=0.06) nor lung function (S1 Table). STST was not correlated with HRQOL items, except for child-estimated social functioning (r=0.37; p=0.03) (S2 Table). Inversely, 6MWD correlated with every muscle strength assessment (r=0.48; r=0.54 and r=0.46; p<0.01 for MIP, MEP and QS respectively) and with lung function (r=0.59; r=0.55 and r=0.58; p<0.01 for FVC, FEV1 and PEF respectively) (S1 Table). 6MWD was associated with several HRQOL items including child physical functioning (S2 Table).”

P17, L254: Results

“Changes in HR during both tests correlated with participant height (r=0.38 and r=0.42; p=0.02 for STST and 6MWT respectively). Delta HR during the STST correlated with lung function (r=0.36, r=0.46 and r=0.50; p<0.05 with FVC, FEV1 and PEF respectively) and with child-reported HRQOL total scores (r=0.46; p<0.01) whereas delta HR during 6MWT correlated with FEV1 and PEF (r=0.35 and r=0.51; p=0.04 and p<0.01 respectively).”

The results of the best test have been added to the appropriate section in Table 1. The correlation coefficients and the p-values associated have also been added in supplemental Tables 1 and 2. Furthermore, the results of the cardio-respiratory adaptations in Table 3 have been replaced by the values achieved during the best attempt of each test. 

P17, L270: Discussion

“This study showed that the STST had moderate criterion validity for the measurement of functional exercise capacity when compared to the 6MWT (r=0.49) in the entire study group of children with CF and in children ≤10 years of age (r=0.54). The STST was not correlated to other outcomes whereas the 6MWT was correlated with muscle strength (respiratory muscles and quadriceps) and HRQOL.”

P19, L311: Discussion

“The mean number of STST repetitions performed in the present study was around 40 (±10).”

P19, L323: Discussion

“Surprisingly, the average 6MWD of 596m in this study was considerably lower than the average distance of 742m reported previously in a sample of children of similar age and lung function [6].”

P20, L335: Discussion

“End-exercise HR during the 6MWT reported in studies including healthy children and children with CF were considerably higher than in the present study with respectively 158 and 154 beats/min compared to 126 beats/min [33, 37].”

P20, L337: Discussion

“Another recent study of healthy children reported a higher peak HR during the STST compared to our study (139 compared to 116 beats/min) [11].”

Sample size

Even though the sample was based on the expected correlation, it is known the validation study has to be, at least, 50 volunteers. 

Thank you for this suggestion. Actually, as the reviewer mentioned, the estimation of the sample size was based to the detection of at least a moderate correlation (r=0.45). This point was also raised by Reviewer#2 and it clearly requires further explanation. Actually, the sample size estimation for the present study was based on a correlation (i.e. criterion validity between two tests) coefficient ‘r’, α (=0.05) and β (usually set at 20% to ensure an 80% power). However, in that particular case, the correlation coefficient estimation can be tricky because a high ‘r’ will lead to a very limited sample size. Two previous studies were initially identified to estimate ‘r’ and perform the present study sample size estimation. The first one (Gurses HN, Zeren M, Denizoglu Kulli H, Durgut E. The relationship of sit-to-stand tests with 6-minute walk test in healthy young adults: Medicine (Baltimore). 2018;97: e9489. doi:10.1097/MD.0000000000009489) reported a correlation coefficient of r=0.647 between STST1’ and 6MWT in healthy young adults. The second one (Radtke T, Puhan MA, Hebestreit H, Kriemler S. The 1-min sit-to-stand test—A simple functional capacity test in cystic fibrosis? J Cyst Fibros. 2016;15: 223–226. doi: 10.1016/j.jcf.2015.08.006) revealed a r=0.627 between STST1’ and VO2peak in adults with CF. 

A sample size estimation based on one of these studies would have been: 

- r=0.647, α=0.05 and β=0.20 → n=16 participants or 21 participants for a 90% power

- r=0.627, α=0.05 and β=0.20 → n=17 participants or 22 participants for a 90% power

According to your comment, this would have been too small a sample size for the purpose of the present study. Therefore, we aimed to perform a sample size estimation to be adequately powered to detect ‘at least’ a moderate correlation between the STST and the 6MWT. In previous publications, a moderate correlation is defined by a ‘r’ of 0.40 or 0.50 (according to Swinscow TDV, Campbell MJ. Statistics at square one. 10., [upgraded] ed. London: BMJ; 2002. 158 p or Hinkle DE, Wiersma W, Jurs SG (2003). Applied Statistics for the Behavioral Sciences 5th ed. Boston: Houghton Mifflin) which is why we aimed to compute a sample size estimation using a moderate ‘r’ value of 0.45 according to previous publications on the same topic (Reychler G, Boucard E, Peran L, Pichon R, Le Ber-Moy C, Ouksel H, Liistro G, Chambellan A, Beaumont M. One-minute sit-to-stand test is an alternative to 6MWT to measure functional exercise performance in COPD patients. Clin Respir J. 2018; 12(3): 1247-1256. doi: 10.1111/crj.12658).

Finally, the sample size estimation was also based on the assumption by both Swinscow and Hinkle, in the previously mentioned references, that a sample size should be at least 30 subjects to investigate criterion validity. Nonetheless, we would be pleased to acknowledge the reference mentioned by the reviewer explaining the need to include 50 subjects and to include this statement as a study limitation if the reviewer wants us to. 

The strength of quadriceps should be included as sample size estimation.

Thank you for this suggestion. We absolutely agree with the reviewer that quadriceps strength is largely associated with functional exercise capacity in the CF population. Previous publications have reported a correlation between STST performance and quadriceps strength in individuals with chronic respiratory disease. Nonetheless, the main aim of the present study was to explore the criterion validity of the STST to measure functional exercise capacity, rather than quadriceps strength, which is why the expected correlation between STST and quadriceps strength was not included in the sample size estimation. 

CPET was assessed? It was not included in the Method, only in the Statistical section.

Thank you for this question. The concerned statement in the ‘Statistical analyses’ section was indeed misleading. CPET was not performed in the present study. This sentence was referring to the latest international statement on exercise testing (Hebestreit et al, Respiration, 2015) that recommends performing CPET or the 6MWT yearly in children with CF from the age of 10. This sentence was used to justify the secondary analysis that classified the CF population into younger (≤10 years old) and older (11-18 years old) children. Comparing STST performance with CPET would have been valuable but CPET was not undertaken in the present study due to equipment unavailability. This has also been pointed out as a limitation of our study. 

P11, L180: Methods

“According to the latest international statement, exercise testing (using either CPET or 6MWT) should be undertaken yearly in all individuals with CF from the age of 10 [2]. We therefore also aimed to investigate STST criterion validity separately in the children ≤ 10 years of age and those aged 11 – 18 years.”

[2] Hebestreit H, Arets HGM, Aurora P, Boas S, Cerny F, Hulzebos EHJ, et al. Statement on Exercise Testing in Cystic Fibrosis. Respiration. 2015;90: 332–351. doi:10.1159/000439057

P18, L284: Discussion

“None of the children in this study underwent CPET before or during our study due to equipment unavailability.”

P21, L371: Discussion

“Future studies may wish to consider assessing the relationship between STST and CPET-derived outcomes in children with CF.”

Statistical Analysis

The Bland Altman plot should be addressed.

Thank you for this suggestion. The following sentence has been added to emphasize the Bland-Altman method plotting in the statistical analysis section. 

P12, L211: Methods

“The Bland-Altman method was used to determine bias and limits of agreement [20].”

[20] Bland JM, Altman DG. Statistical methods for assessing agreement between two methods of clinical measurement. Lancet Lond Engl. 1986;1: 307–310.

Results

Quality of life, Borg, blood pressure, respiratory rate (...) were not described in the Method Section.

Thank you for this comment. According to one of your previous comments, the following outcome assessments have been specified in the methods section. The following changes have been made. 

P9, L142: Methods

“Heart rate (HR), pulsed oxygen saturation (SpO2), systolic and diastolic blood pressure (SBP, DBP), and respiratory rate (RR) were measured during a 1-min sitting rest period before and after each STST (53NOP portable monitor, WelchAllyn). Dyspnea and lower limb fatigue were also recorded using the modified Borg scale.”

P10, L151: Methods

“HR, SpO2, SBP, DBP, RR, dyspnea and lower limb fatigue were also recorded during a 1-min sitting rest period before and after each 6MWT.”

P11, L174: Methods

“Functional testing was then undertaken according to the protocol described above and health-related quality of life (HRQOL) was evaluated using the French version of the CFQ-R during the rest period between two iterations of the field tests [16]. For children under 14 years of age, a parent also completed the parental version of the CFQ-R.”

Table 1 is usually used to describe the population characteristics. The outcomes should be in table 2.

Thank you for this suggestion. Table 1 has been shortened to show only the population characteristics and muscle and functional outcomes have been moved to a new Table 2. 

There was follow up? Why it is written in the flow diagram?

Thank you for this question. The reviewer is right and this was a reporting mistake in the CONSORT Flow diagram. There was indeed no ‘follow-up’ in the present study. This section has been removed from the Figure 1 (flow diagram). 

Discussion and Conclusion

In my believe, it is not possible to conclude that STST is not valid test! The sample size is not enough to guarantee it. STST has different information than 6MWT, specially for the test time. Thus, the authors should not expect to change on for other.

Thank you for this comment. This concern was also raised by Reviewer#2 and we agree that the primary version of the discussion and conclusion was misleading. Indeed, the tests are not interchangeable and the purpose of the present study was not to propose the STST as a complete surrogate to the 6MWT. We have now highlighted the different aspects of functionality measured by both tests, the differences in their durations and have modified the clinical message of the present study according to your comments. 

P18, L285: Discussion

“This moderate correlation can be explained by the different domains of physical capacity investigated by both tests, and the corresponding differences in energy expenditure [22,23]. The sit to stand movement is relevant for activities of daily living and is repeated frequently throughout the day by both adults and children. The lack of ability of the one-minute STST to predict the distance covered during a 6MWT can be attributed to the functional differences between the tests, as well as differences in energy utilization between walking for 6-minutes and standing repeatedly from a chair for 1-minute. Interestingly, a longer variation of the one-minute STST (the 3-minute STST) induced a similar functional response to that of the 6MWT in patients with COPD, leading the authors to conclude that the tests were equivalent [24]. Nevertheless, it should be noted that the 3-minute STST induced a higher degree of leg fatigue than the 6MWT, reinforcing the suggestion that the tests have different functional demands and should not be used interchangeably. The STST remains useful if the 6MWT cannot be performed.”

References

[22] Ridley K, Olds TS. Assigning Energy Costs to Activities in Children: A Review and Synthesis. Med Sci Sports Exerc. 2008;40: 1439–1446. doi:10.1249/MSS.0b013e31817279ef

[23] Butte NF, Watson KB, Ridley K, Zakeri IF, McMurray RG, Pfeiffer KA, et al. A Youth Compendium of Physical Activities: Activity Codes and Metabolic Intensities. Med Sci Sports Exerc. 2018;50: 246–256. doi:10.1249/MSS.0000000000001430

P20, L341: Discussion

“This difference in cardiorespiratory adaptation, which has previously been demonstrated in healthy adults performing the same tests, may be due to the different durations of each test [27]. Greater adaptation could be expected for a six-minute walk compared to a task like standing repeatedly from a chair for only one minute. Furthermore, walking could involve a larger number of muscles, inducing a faster HR and a higher level of leg fatigue and dyspnea. These differences further strengthen the hypothesis that these tests measure different aspects of functional capacity.”

P21, L362: Discussion

“Furthermore, it is important to note that the STST does not measure the same aspects of functional capacity as the 6MWT, and that its discriminative validity as well as its clinical applications (e.g. the association with hospitalizations for CF or other important prognostic outcomes) have yet to be demonstrated.”

P22, L377: Conclusion

“The results of this study suggest the STST measures different aspects of functional capacity than the 6MWT in children with CF. STST performance was moderately correlated with 6MWD, and the test induced a lower cardio-respiratory response. Neither quadriceps nor respiratory muscle strength were related to the number of STST repetitions, and the clinical value of the test is still to be proven. The simplicity of the STST for both children and practitioners makes it a reasonable second option when 6MWT or CPET cannot be performed, but healthcare providers should be aware that these tests cannot be used interchangeably.”

---

## [Decision Letter · Decision Letter 1]

19 Nov 2020

PONE-D-20-11696R1

Measurement properties of the one-minute sit-to-stand test in children and adolescents with cystic fibrosis: a multicenter randomized cross-over trial

PLOS ONE

Dear Dr. Combret,

Thank you for submitting your manuscript to PLOS ONE. After careful consideration, we feel that it has merit but does not fully meet PLOS ONE’s publication criteria as it currently stands. Therefore, we invite you to submit a revised version of the manuscript that addresses the points raised during the review process.

Please take note and amend the last minor notes highlighted by the reviewer n°2

We look forward to receiving your revised manuscript.

Kind regards,

Andrea Martinuzzi

Academic Editor

PLOS ONE

Reviewers' comments:

Reviewer's Responses to Questions

**Comments to the Author**

1. If the authors have adequately addressed your comments raised in a previous round of review and you feel that this manuscript is now acceptable for publication, you may indicate that here to bypass the “Comments to the Author” section, enter your conflict of interest statement in the “Confidential to Editor” section, and submit your "Accept" recommendation.

Reviewer #1: All comments have been addressed

Reviewer #2: All comments have been addressed

Reviewer #3: All comments have been addressed

2. Is the manuscript technically sound, and do the data support the conclusions?

Reviewer #1: Yes

Reviewer #2: Yes

Reviewer #3: Yes

3. Has the statistical analysis been performed appropriately and rigorously? 

Reviewer #1: Yes

Reviewer #2: Yes

Reviewer #3: Yes

4. Have the authors made all data underlying the findings in their manuscript fully available?

Reviewer #1: Yes

Reviewer #2: Yes

Reviewer #3: Yes

5. Is the manuscript presented in an intelligible fashion and written in standard English?

Reviewer #1: Yes

Reviewer #2: Yes

Reviewer #3: Yes

6. Review Comments to the Author

Reviewer #1: Since the comments made on earlier draft by me (and hopefully by other respected reviewers also) are attended positively/adequately, now the manuscript is improved a lot. No major issue left, in my opinion.

Reviewer #2: The authors have provided sufficient justification to my queries and have addressed most of my concerns. The manuscript improved substantially. However, I would like to suggest them to review some other aspects.

1- Line 200, Page 200: To investigate criterion validity, it is recommended to use tests that have the same characteristics and objective, and ideally are the gold standard. The authors should specify which outcomes were used to investigate criterion validity.

2- Although learning effect and intra-rater reliability was performed, they were not presented in study’s conclusion. Since validity has not been stablished, the authors should describe these results at the conclusion to support the use of 1STST.

3- Although moderate correlations were found for both analysis with the entire sample and with children≤10 years of age, the result with children≤10 years seems to have higher correlation. Was there a correlation with the sample of children>10 years age? The average age of the sample is 12 years, perhaps the characteristics of the sample have influenced the results. The authors should discuss the result in the correlation analysis with children<10 years age.

4- In line with item 3, to add information to the reader, the authors could describe in table 1 and table 2 the characteristics of the children≤10 years age.

Reviewer #3: All the suggestions were accepted by the authors and the doubts were answered. The sample size was better described, and the best test was now used for the analysis.

The authors were correct to better described the method and to described the outcomes, that were not performed before.

7. PLOS authors have the option to publish the peer review history of their article (what does this mean?). If published, this will include your full peer review and any attached files.

Reviewer #1: **Yes: **Dr. Sanjeev Sarmukaddam

Reviewer #2: **Yes: **Camile Ludovico Zamboti

Reviewer #3: **Yes: **Fernanda C. Lanza

---

## [Author Response · Author response to Decision Letter 1]

28 Nov 2020

From: Yann Combret, 27th November 2020

To: Dr. Andrea Martinuzzi

Academic Editor, Plos One

Dear Dr. Martinuzzi, 

Thank you for your invitation to provide a revised version of our work. We are sincerely grateful to you and the reviewers for the time you spent reviewing our paper entitled “Measurement properties of the one-minute sit-to-stand test in children and adolescents with cystic fibrosis: a multicenter randomized cross-over trial”. We are pleased that the reviewers appreciated the corrections made after their suggestions. The other aspects outlined by the Reviewer#2 have been addressed in this revised version. The following is a point-by-point response to the reviewers’ comments. We have provided a marked-up version of our manuscript and a clean version along with this response according to your recommendations. We hope that you will find the responses and revisions satisfactory and that the manuscript now meets Plos One publication criteria.

 We look forward to hearing from you in the near future.

 Yours sincerely.

Yann COMBRET and Gregory REYCHLER

Please find below our point-by-point response to the reviewers (with modifications highlighted).

 

Response to Reviewer #2

The authors have provided sufficient justification to my queries and have addressed most of my concerns. The manuscript improved substantially. However, I would like to suggest them to review some other aspects.

Thank you for this positive comment and your helpful suggestions. The following is a point-by-point response to your new comments. 

1- Line 200, Page 200: To investigate criterion validity, it is recommended to use tests that have the same characteristics and objective, and ideally are the gold standard. The authors should specify which outcomes were used to investigate criterion validity.

Thank you for this comment. Criterion validity of the STST to measure functional exercise capacity was measured though the correlation between STST performance and the 6MWD. The gold standard for exercise testing in CF patients is indeed laboratory exercise testing using cycle or treadmill CPET; but, the 6MWT is the most widely used field test worldwide and should be considered whenever CPET is unachievable (Hebestreit H, Arets HGM, Aurora P, Boas S, Cerny F, Hulzebos EHJ, et al. Statement on Exercise Testing in Cystic Fibrosis. Respiration. 2015;90: 332–351). 

The measurement undertaken was precised at the end of the ‘Introduction’ (see Line 103) and is now further described in the ‘Methods’ according to the reviewer comment.

P12, L201: “Criterion validity of the STST to measure functional exercise capacity was explored by measuring the correlation between STST performance and the 6MWD derived from the reference test performance (i.e. 6MWT), using Spearman or Pearson correlation coefficients in the complete study sample, in children ≤10 years of age and in children aged 11-18 years.”

2- Although learning effect and intra-rater reliability was performed, they were not presented in study’s conclusion. Since validity has not been stablished, the authors should describe these results at the conclusion to support the use of 1STST.

Thank you for this suggestion. The ‘Conclusion’ has been rephrased to encompass these results. 

P23, L394: “The simplicity of the STST for both children and practitioners, and the very good intra-rater reliability reported in this study, makes it a reasonable second option when 6MWT or CPET cannot be performed, but healthcare providers should be aware that these tests cannot be used interchangeably. Nonetheless, as both tests revealed a significant learning effect that should be taken to account, the STST could be helpful to reduce the duration of exercise testing.”

3- Although moderate correlations were found for both analysis with the entire sample and with children≤10 years of age, the result with children≤10 years seems to have higher correlation. Was there a correlation with the sample of children>10 years age? The average age of the sample is 12 years, perhaps the characteristics of the sample have influenced the results. The authors should discuss the result in the correlation analysis with children<10 years age.

Thank you for these suggestions. The correlation between STST and 6MWD was added for children >10 years of age in the ‘Results’ section of the manuscript. As the reviewer will notice, the correlation is significant; but, lower than the one of younger children (r=0.43; p=0.04). Indeed, as the reviewer hypothesized, the characteristics of the subgroups could be responsible for this difference. Hence, both STST and 6MWT performance are linked with sex; and the sex ratio was different between both subgroups (Reychler G, Cabillic M, Morales Mestre N, Poncin W, Audag N, Caty G. Predictive model for the 1-minute sit-to-stand test in healthy children aged 6 to 12 years. Ann Phys Rehabil Med. 2020; Li AM, Yin J, Au JT, So HK, Tsang T, Wong E, et al. Standard Reference for the Six-Minute-Walk Test in Healthy Children Aged 7 to 16 Years. Am J Respir Crit Care Med. 2007;176: 174–180). There was 64% of boys in the ‘younger children’ group, and 23% of boys in the ‘older children’ group. One hypothesis is that in the ‘younger children’ group, the sex ratio was almost equivalent, leading to more heterogenous performances which could have strengthened the correlation between both tests (Swinscow TDV, Campbell MJ. Statistics at square two. 10., [upgraded] ed. London: BMJ; 2002). On the opposite, the unbalanced ratio in the ‘older children’ group could have decreased the variability among performances, thereby lowering the correlation coefficient calculated. 6MWD was also more heterogenous in the ‘younger children group’ (i.e. higher SD), probably because this group was smaller (14 children vs 22 in the ‘older group’). Notwithstanding, the difference in correlation coefficient is small, and both correlations could be labeled as ‘moderate’ and this result requires confirmation from larger cohorts.

P20, L314: “Finally, a small trend for a higher correlation between STST and 6MWT performance in children ≤ 10 years of age, compared to older children (>10 years of age) was observed in this study (r=0.54 vs r=0.43, respectively). Interestingly, a sex effect on both STST and 6MWT performance has previously been described; and the sex ratio was unequally distributed between those subgroups (64 vs 23% of boys between the children ≤ 10 years of age and >10 years of age, respectively) [32,33]. A higher heterogeneity in both tests performances in the group of children ≤ 10 years of age could thereby have led to a higher correlation coefficient [18]. Therefore, the STST could be a promising tool for exercise testing in young children with CF, but confirmation in a larger sample of participants is needed.”

4- In line with item 3, to add information to the reader, the authors could describe in table 1 and table 2 the characteristics of the children≤10 years age.

Thank you for this helpful proposal. The characteristics of the children ≤ 10 years of age (and subsequently those of the children aged 11-18 years) are now detailed in Table 1 and 2.

---

## [Decision Letter · Decision Letter 2]

27 Jan 2021

Measurement properties of the one-minute sit-to-stand test in children and adolescents with cystic fibrosis: a multicenter randomized cross-over trial

PONE-D-20-11696R2

Dear Dr. Combret,

We’re pleased to inform you that your manuscript has been judged scientifically suitable for publication and will be formally accepted for publication once it meets all outstanding technical requirements.

Kind regards,

Andrea Martinuzzi

Academic Editor

PLOS ONE

Additional Editor Comments (optional):

Reviewers' comments:

Reviewer's Responses to Questions

**Comments to the Author**

1. If the authors have adequately addressed your comments raised in a previous round of review and you feel that this manuscript is now acceptable for publication, you may indicate that here to bypass the “Comments to the Author” section, enter your conflict of interest statement in the “Confidential to Editor” section, and submit your "Accept" recommendation.

Reviewer #1: All comments have been addressed

Reviewer #2: All comments have been addressed

Reviewer #3: All comments have been addressed

2. Is the manuscript technically sound, and do the data support the conclusions?

Reviewer #1: Yes

Reviewer #2: Yes

Reviewer #3: Yes

3. Has the statistical analysis been performed appropriately and rigorously? 

Reviewer #1: Yes

Reviewer #2: Yes

Reviewer #3: Yes

4. Have the authors made all data underlying the findings in their manuscript fully available?

Reviewer #1: Yes

Reviewer #2: Yes

Reviewer #3: Yes

5. Is the manuscript presented in an intelligible fashion and written in standard English?

Reviewer #1: Yes

Reviewer #2: Yes

Reviewer #3: Yes

6. Review Comments to the Author

Reviewer #1: As said earlier, all the comments were already attended positively/adequately, now the manuscript is improved a lot. No major issue left, in my opinion.

Reviewer #2: The manuscript improved substantially and the authors have provided justification to all my queries.

Reviewer #3: All the suggestions were accepted by the authors. I am happy to tell that, for me, the paper is well written and can be accepted.

7. PLOS authors have the option to publish the peer review history of their article (what does this mean?). If published, this will include your full peer review and any attached files.

Reviewer #1: **Yes: **Dr. Sanjeev Sarmukaddam

Reviewer #2: No

Reviewer #3: **Yes: **Fernanda C Lanza

---

## [Editor Report · Acceptance letter]

3 Feb 2021

PONE-D-20-11696R2 

Measurement properties of the one-minute sit-to-stand test in children and adolescents with cystic fibrosis: a multicenter randomized cross-over trial 

Dear Dr. Combret:

I'm pleased to inform you that your manuscript has been deemed suitable for publication in PLOS ONE. Congratulations! Your manuscript is now with our production department. 

Kind regards, 

on behalf of

Dr. Andrea Martinuzzi 

Academic Editor

PLOS ONE